# Discovering two general characteristic times of transient responses in solid oxide cells

Zhaojian Liang [1], Jingyi Wang [2], Keda Ren[2], Zhenjun Jiao[2], Meng Ni [3], Liang An [1], Yang Wang[3], Jinbin Yang[2] & Mengying Li [1] ✉

A comprehensive understanding of the transient characteristics in solid oxide cells (SOCs) is crucial for advancing SOC technology in renewable energy storage and conversion. However, general formulas describing the relationship between SOC transients and multiple parameters remain elusive. Through comprehensive numerical analysis, we find that the thermal and gaseous response times of SOCs upon rapid electrical variations are on the order of two characteristic times ($\tau_h$ and $\tau_m$), respectively. The gaseous response time is approximately $1\tau_m$, and the thermal response time aligns with roughly $2\tau_h$. These characteristic times represent the overall heat and mass transfer rates within the cell, and their mathematical relationships with various SOC design and operating parameters are revealed. Validation of $\tau_h$ and $\tau_m$ is achieved through comparison with an in-house experiment and existing literature data, achieving the same order of magnitude for a wide range of electrochemical cells, showcasing their potential use for characterizing transient behaviors in a wide range of electrochemical cells. Moreover, two examples are presented to demonstrate how these characteristic times can streamline SOC design and control without the need for complex numerical simulations, thus offering valuable insights and tools for enhancing the efficiency and durability of electrochemical cells.

High penetration of intermittent solar and wind power can adversely affect the stability and safety of electrical grids. As a high-temperature electrochemical device using cost-effective metals and earth-abundant catalysts, Solid Oxide Cell (SOC) offers economic advantages for large-scale renewable energy storage[1,2]. SOCs can operate reversibly in both electrolysis and fuel cell modes[3,4]. In electrolysis mode, solid oxide electrolysis cells (SOECs) can convert the intermittent solar[5] and wind energy[6] to the chemical energy of alternative fuels such as hydrogen and syngas[7-10]. In fuel cell mode, solid oxide fuel cells (SOFCs) can generate power from fuels previously produced by SOECs[11,12]. Due to its high flexibility and efficiency[7,11,13,14], SOC technology has immense potential in mitigating the intermittency challenges associated with renewable power generation.

When integrated with intermittent power sources, SOCs may frequently operate under fluctuating electrical conditions. The electrical response time of SOC is constrained by the slow heat and mass transfer processes within the cell[15,16]. Varying load changes may result in under-performance, thermal safety concerns, and degradation of SOCs[11,17-19]. To mitigate these challenges, researchers have proposed several control and design strategies[20-24] aimed at facilitating safe operation and improving the inherent transient characteristics of SOCs. For example, Nerat[19] recommended maintaining the anode supporting layer thickness above 0.1 mm to prevent local fuel starvation during the dynamic operation of a planar SOFC, and proposed a load variation rate with a time constant of 50 ms to minimize electrical overshoot. Bae et al.[25] emphasized the significance of fuel utilization on

[1]Department of Mechanical Engineering & Research Institute for Smart Energy, The Hong Kong Polytechnic University, Hong Kong, Hong Kong SAR. [2]School of Science, Harbin Institute of Technology, Shenzhen, China. [3]Department of Building and Real Estate, The Hong Kong Polytechnic University, Hong Kong, Hong Kong SAR. ✉e-mail: mengying.li@polyu.edu.hk

the current relaxation time of SOFCs following electrical load changes. While these studies offer valuable insights into SOC transient behaviors, their quantitative findings are seldom applicable to other SOCs due to the diverse cell specifications and operating conditions. Therefore, optimized design parameters or control strategies for one specific SOC may not be effective for another SOC. Essentially, the diverse nature of transient phenomena in SOCs limits the generalizability of control strategies and design principles for these systems.

To establish a generalized principle to quantify the transient characteristics of SOCs, a few efforts have been made to identify a characteristic time constant that governs the SOC transient behavior. Some studies[19,26–28] proposed that SOC electrical responses depend on the time constants of heat and mass diffusion in the thickness direction of the SOC, namely, $\delta^2/\alpha$ and $\delta^2/D$. However, this perspective does not account for the significant effects of cell length[29] and inlet flow rates[20,25] on SOC transients. In contrast, other works[15,30] employed the convective time constant $L_{cell}/V_{in}$ to characterize SOC transients, but this approach does not adequately explain the substantial impacts of electrode thickness and porosity[19,25]. In summary, the existing research does not adequately describe the relationship between SOC transients and their associated design and operating parameters. The absence of a generalized expression necessitates substantial computational and experimental efforts to characterize the transient behaviors of SOCs. This research gap hinders the optimizations of SOC design and control strategies, as well as the application of SOCs in renewable energy storage and conversion.

To address the prevailing research gap, the present study develops a distinctive theoretical framework for the investigation of transient behaviors of SOCs. We uncover generalized and straightforward mathematical formulas to calculate characteristic times that govern the transport phenomena within the SOCs. Our findings offer innovative insights into the correlation between transient characteristics of SOCs and various parameters, thereby enabling more effective design and control of SOCs at greatly reduced computational and experimental efforts. Moreover, the incorporation of non-dimensional analysis within SOCs may stimulate new comprehension of transient behavior in electrochemical cells with analogous structures and flow phenomena. Such cells encompass proton exchange membrane fuel cells (PEMFCs) and protonic ceramic fuel cells/electrolyzers.

## Results

In SOEC operations, electrochemical reactions utilize electricity to dissociate steam into hydrogen and oxygen. Conversely, during SOFC operations, hydrogen and oxygen are consumed to produce steam and electricity. The general working principle and structure of SOEC are depicted in Fig. 1a. Both the top and bottom interconnects function as electron conductors. The porous media, including the Anode Diffusion Layer (ADL), Anode Functional Layer (AFL), Cathode Functional Layer (CFL), and Cathode Diffusion Layer (CDL), facilitate electron conduction and contain pores for gas transport. At high temperatures, oxygen ion transport occurs predominantly within the solid oxide electrolyte, AFL, and CFL. In SOEC mode, heated steam from the fuel channel diffuses through the CDL to the CFL, where cathodic reactions take place at the triple-phase or double-phase boundaries, resulting in the dissociation of steam into hydrogen and oxygen ions. Subsequently, oxygen ions traverse the solid oxide electrolyte to the AFL, where anodic reactions generate oxygen. Our preceding study[15] demonstrated that the transient characteristics of SOCs are significantly influenced by the transport processes of electrons, ions, gases, and heat.

### Characteristic time of mass transfer

To understand the transient mass-transfer behavior of SOCs (including both SOEC and SOFC), we manipulate parameters that directly affect mass transfer rates and investigate how the mass field responds to changing current conditions with different parameter values. Since the current directly reflects the generation or consumption of species in FL, the transient response of species can be interpreted as the gaseous response after a step change of mass source. Table 1 presents the values of the influencing parameters of 27 studied cases, including one base case. The cells in cases 1 to 23 operate under SOEC mode, while the remaining cases operate under SOFC mode. Each case has a couple of parameters that differ from the base case but follows the same simulation procedure as outlined below,

1. Each case is initialized to the steady-state solution corresponding to $i = i_{t=0}$.
2. The external current applied on the SOC changes from $i_{t=0}$ to $i_{t>0}$ within a small time step ($<10^{-5}$ s) and is held at $i_{t>0}$ until the end of simulation. For all the cases, $i_{t=0}$ and $i_{t>0}$ are within the limiting current $i_{lim}$, which serves as the theoretical maximum current density achievable under a given reactant supply.
3. The SOC temperature is constant during the simulation to exclude the effect of temperature variation on mass transfer.
4. The electrical behaviors and average mole fractions of species in FLs are monitored throughout the simulation.

The transient simulation results of SOEC cases are illustrated in Fig. 2a–c. The temporal variations of voltage $U$ and mole fractions of water $X_{H_2O}$ and oxygen $X_{O_2}$ are presented in Fig. 2a. However, direct comparison of transients across different cases is challenging as all cases have distinguished initial and final steady states. To facilitate comparison between cases, $U$, $X_{H_2O}$, and $X_{O_2}$ are scaled to $U^*$, $X_{H_2O}^*$, and $X_{O_2}^*$, respectively, using the scales presented in the caption of Fig. 2. The values of $U^*$, $X_{H_2O}^*$, and $X_{O_2}^*$ represent the extent of steadiness achieved, with the value of 1 indicating 100% reaching the new steady state. In this study, the time required for a variable to reach 95% of its steady-state value (e.g. $X_{H_2O}^* = 0.95$) is defined as the relaxation time. The plots of $U^*$, $X_{H_2O}^*$, and $X_{O_2}^*$ with respect to time (as shown in Fig. 2b) reveal that nearly all the studied parameters affect the relaxation time.

In order to elucidate the general relationship between multiple parameters and the relaxation time, we investigated a range of time constants, including the diffusive time constant in the thickness direction, $\delta^2/D$[19,28], and the convective time constant, $L_{cell}/V_{in}$[15,30]. While both diffusive and convective time constants are inherently used to describe 1-D local transport phenomena, the time constants for 1-D transport cannot fully describe the 3-D transport phenomena within SOCs. To generalize the 3-D transport phenomena, we apply the concept of control volume[31] from Fluid Mechanics to the SOC. Control volume is a closed region defined in space, which is utilized to focus our attention on the mass and energy crossing the boundary and the conservation law within the region[31]. Either the fuel side or air side of the SOC can be considered as a control volume. Take the fuel side as an example, the inlet and outlet of fuel channel allow the crossing of fuel flow, while the other boundaries of the control volume are impermeable. The total mass of the control volume obeys the conservation law. By using the non-dimensional analysis on the mass conservation equation of the control volume, we derived the mass-transfer characteristic time, $\tau_m$, for SOCs. The detailed derivation is presented in Supplementary Methods. The expression for $\tau_m$, as presented in Eq. (1), reflects the time required for the fluid to flow through the void volume in SOC.

$$\tau_m = \frac{\text{Void volume}}{\text{Volumetric flow rate}}\frac{[\text{m}^3]}{[\text{m}^3\,\text{s}^{-1}]} = \frac{\mathcal{V}}{\dot{\mathcal{V}}_0}$$

$$= \begin{cases} \dfrac{W_{ch}H_{ch}L_{cell} + W_{cell}L_{cell}\left(\varepsilon_{DL}^{fuel}\delta_{DL}^{fuel} + \varepsilon_{FL}^{fuel}\delta_{FL}^{fuel}\right)}{W_{ch}H_{ch}V_{in}^{fuel}} & \text{Fuel side} \\[2em] \dfrac{W_{ch}H_{ch}L_{cell} + W_{cell}L_{cell}\left(\varepsilon_{DL}^{air}\delta_{DL}^{air} + \varepsilon_{FL}^{air}\delta_{FL}^{air}\right)}{W_{ch}H_{ch}V_{in}^{air}} & \text{Air side} \end{cases} \quad (1)$$

By scaling the time with the $\tau_m$ of each case, Fig. 2c is obtained, where the relaxation times of all cases converge to one point. The

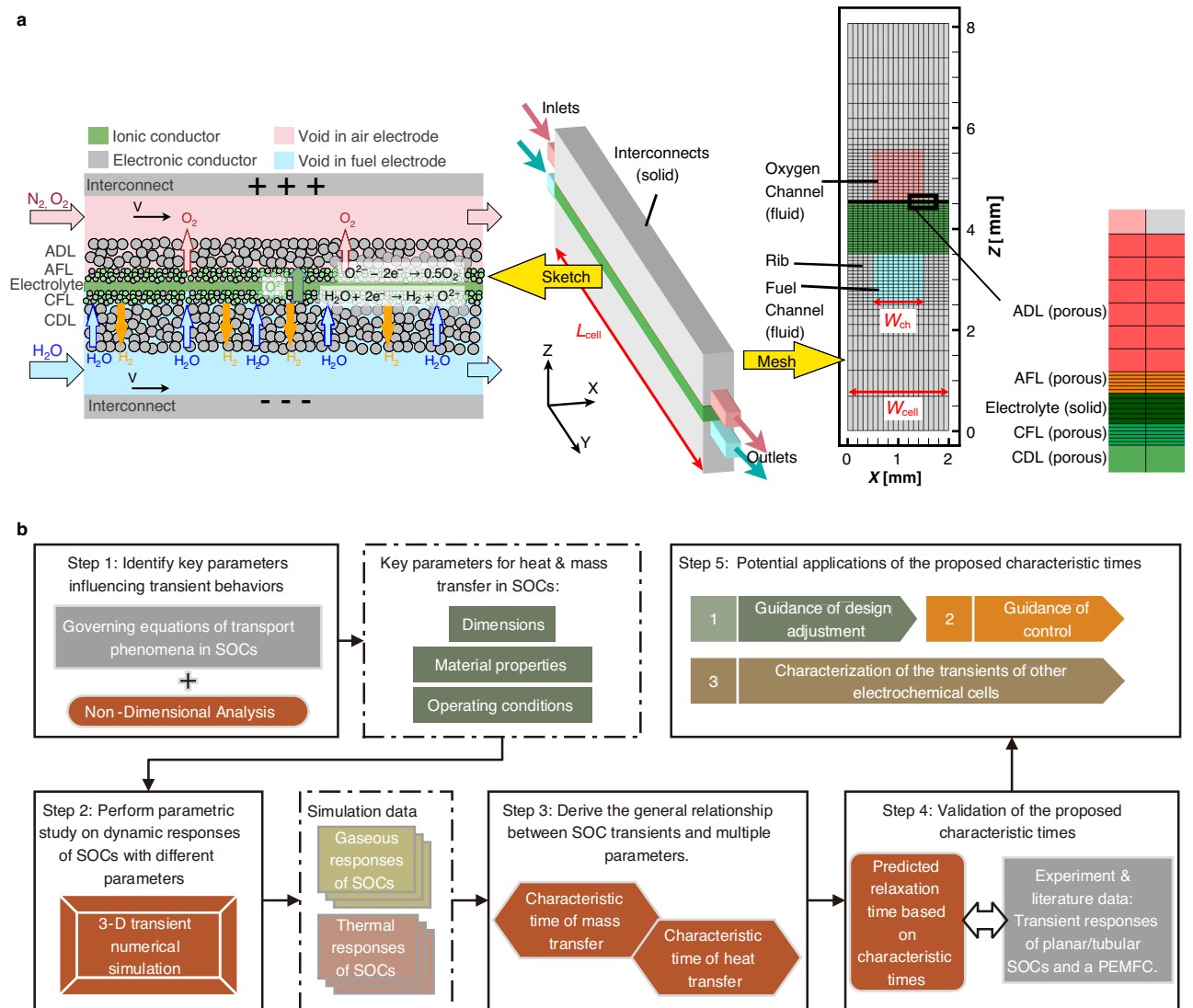

**Fig. 1 | The methodology employed in this study. a** The modeling geometry and working principle of SOEC. **b** The overall methodology employed in this work.

results show that $X^*_{H_2O}$ in the fuel side and $X^*_{O_2}$ in the air side take around $1\tau_m^{fuel}$ and $1\tau_m^{air}$, respectively, to stabilize after a step change of current. In additions, the relaxation time of $U^*$ is approximately $1\tau_m^{fuel}$.

Note that, according to Eq. (1), $\tau_m$ may have different values on the fuel side and the air side of SOC due to the differences in flow conditions and electrode structures. This observation is consistent with Nerat's study[19], which claimed that the gaseous response in the fuel electrode is slower than that in the air electrode due to the significantly thicker diffusion layer (DL) in the fuel side.

Our investigation is extended from SOECs to SOFCs, as shown in Fig. 2d–f. The operating conditions for SOECs and SOFCs differ significantly, as presented in Table 1. For example, SOFCs require large air flow rates to evacuate the heat generated by the electrochemical reactions[32]. Figure 2d,e present the complex interplay between SOFC transients and various parameters, highlighting the contrast between SOEC and SOFC transients. In Fig. 2f, time is scaled with $\tau_m$. The relaxation time of $U^*$ is found to be approximately $1\tau_m^{fuel}$. Interestingly, if using $\tau_m^{air}$ as time scale, the relaxation time of $U^*$ would span a range from 2 to 20 $\tau_m^{air}$ (see Supplementary Fig. 4). This suggests that the species transport in the fuel electrode is the limiting factor of the electrical response for SOCs. Besides, the relaxation times for $H_2$ and $O_2$ are around $1\tau_m^{fuel}$ and $1\tau_m^{air}$ respectively, after the step change of current. This finding further proves the effectiveness of $\tau_m$ in

characterizing mass-transfer transients for SOCs (including both SOECs and SOFCs).

## Characteristic time of heat transfer

Temperature exerts a profound impact on SOC performance by affecting various properties such as the ionic conductivity of the electrolyte and the activation loss of electrochemical reactions[33]. Consequently, heat transfer within the SOC plays a crucial role in determining its transient performance. Under SOEC mode, the cell is endothermic below the thermal neutral voltage and exothermic beyond it. While under SOFC mode, the cell is exothermic. The heat sources in SOEC and SOFC vary with both voltage and current, which are affected by both mass and heat transfer characteristics. To eliminate the influence of mass transfer and other factors on heat sources, we have introduced the following strategy to investigate the characteristic time of heat transfer in SOC, while detailed justifications of this strategy are provided in Supplementary Methods:

- The heat source in SOC is determined independently according to typical SOC operations instead of being calculated based on current and voltage. While our investigation primarily focuses on the step change of the heat source, the results derived from this step change can serve as a reasonable approximation for the corresponding step change in the electrical state.

**Table 1 | Manipulated variables used in the parametric study of mass transfer ('-' indicates that the parameter is consistent with the base case)**

| Name | Case | $\delta_{FL}^{air}$ [m] | $\delta_{FL}^{fuel}$ [m] | $\delta_{DL}^{fuel}$ [m] | $\delta_{DL}^{air}$ [m] | $H_{ch}$ [m] | $W_{ch}$ [m] | $W_{cell}$ [m] | $L_{cell}$ [m] | $\varepsilon_{DL}^{fuel}$ | $\varepsilon_{FL}^{fuel}$ | $\varepsilon_{DL}^{air}$ | $\varepsilon_{FL}^{air}$ | $i_{r=O}$ [A cm⁻²] | $i_{r>O}$ [A cm⁻²] | $T_{in}$ [K] | $X_{in}^{H_2O}$ | $X_{in}^{H_2}$ | $V_{in}^{air}$ [m s⁻¹] | $V_{in}^{fuel}$ [m s⁻¹] | $i_{lim}^{a}$ [A cm⁻²] | Power$^b$ [W cm⁻²] |
|---|---|---|---|---|---|---|---|---|---|---|---|---|---|---|---|---|---|---|---|---|---|---|
| Base | 1 | 7.0E-06 | 7.0E-06 | 1.0E-03 | 4.5E-05 | 1.0E-03 | 1.0E-03 | 2.0E-03 | 0.10 | 0.38 | 0.20 | 0.27 | 0.27 | -1.00 | -0.50 | 1073.15 | 0.70 | 0.30 | 2.00 | 2.00 | -1.53 | -1.08 → -0.50 |
| δ1 | 2 | 1.4E-05 | 1.4E-05 | - | - | - | - | - | - | - | - | - | - | - | - | - | - | - | - | - | - | -1.07 → -0.49 |
| δ2 | 3 | - | - | 5.0E-04 | - | - | - | - | - | - | - | - | - | - | - | - | - | - | - | - | - | -1.08 → -0.50 |
| δ3 | 4 | - | - | - | 9.0E-05 | - | - | - | - | - | - | - | - | - | - | - | - | - | - | - | - | -1.08 → -0.50 |
| H1 | 5 | - | - | - | - | 2.0E-03 | - | - | - | - | - | - | - | - | - | - | - | - | - | - | -3.07 | -1.06 → -0.49 |
| W1 | 6 | - | - | - | - | - | 5.0E-04 | 1.0E-03 | - | - | - | - | - | - | - | - | - | - | - | - | - | -1.08 → -0.50 |
| L1 | 7 | - | - | - | - | - | - | - | 0.05 | - | - | - | - | - | - | - | - | - | - | - | -3.07 | -1.06 → -0.49 |
| ε1 | 8 | - | - | - | - | - | - | - | - | 0.76 | - | - | - | - | - | - | - | - | - | - | -1.61 | -1.09 → -0.50 |
| ε2 | 9 | - | - | - | - | - | - | - | - | - | 0.40 | - | - | - | - | - | - | - | - | - | -1.47 | -1.08 → -0.50 |
| ε3 | 10 | - | - | - | - | - | - | - | - | - | - | 0.54 | - | - | - | - | - | - | - | - | -1.10 | -1.09 → -0.50 |
| ε4 | 11 | - | - | - | - | - | - | - | - | - | - | - | 0.54 | - | - | - | - | - | - | - | -1.97 | -1.08 → -0.50 |
| I1 | 12 | - | - | - | - | - | - | - | - | - | - | - | - | - | -0.75 | - | - | - | - | - | - | -1.08 → -0.78 |
| I2 | 13 | - | - | - | - | - | - | - | - | - | - | - | - | - | -1.25 | - | - | - | - | - | - | -1.08 → -1.41 |
| T1 | 14 | - | - | - | - | - | - | - | - | - | - | - | - | - | - | 1023.15 | - | - | - | - | - | -1.17 → -0.52 |
| T2 | 15 | - | - | - | - | - | - | - | - | - | - | - | - | - | - | 1123.15 | - | - | - | - | - | -1.03 → -0.48 |
| X1 | 16 | - | - | - | - | - | - | - | - | - | - | - | - | - | - | - | 0.50 | 0.50 | - | - | - | -1.14 → -0.52 |
| X2 | 17 | - | - | - | - | - | - | - | - | - | - | - | - | - | - | - | 0.90 | 0.10 | - | - | - | -1.04 → -0.48 |
| V1 | 18 | - | - | - | - | - | - | - | - | - | - | - | - | - | - | - | - | - | 1.50 | 1.50 | -1.15 | -1.11 → -0.50 |
| V2 | 19 | - | - | - | - | - | - | - | - | - | - | - | - | - | - | - | - | - | 2.50 | 2.50 | -1.92 | -1.07 → -0.49 |
| V3 | 20 | - | - | - | - | - | - | - | - | - | - | - | - | - | - | - | - | - | 4.00 | 4.00 | -3.07 | -1.06 → -0.49 |
| VX1 | 21 | - | - | - | - | - | - | - | - | - | - | - | - | - | - | - | 1.00 | 0.00 | 1.00 | 1.00 | -1.10 | -1.08 → -0.48 |
| VH1 | 22 | - | - | - | - | 2.0E-03 | - | - | - | - | - | - | - | - | - | - | - | - | 1.00 | 1.00 | - | -1.09 → -0.50 |
| VH2 | 23 | - | - | - | - | 5.0E-04 | - | - | - | - | - | - | - | - | - | - | - | - | 4.00 | 4.00 | - | -1.08 → -0.50 |
| FC | 24 | - | - | - | - | - | - | - | - | - | - | - | - | 1.50 | 0.50 | - | 0.10 | 0.90 | 18.00 | 2.00 | 1.97 | 1.11 → 0.47 |
| FC-T | 25 | - | - | - | - | - | - | - | - | - | - | - | - | 1.50 | 0.50 | 1123.15 | 0.10 | 0.90 | 18.00 | 2.00 | 1.88 | 1.16 → 0.47 |
| FC-I | 26 | - | - | - | - | - | - | - | - | - | - | - | - | 1.50 | 1.00 | - | 0.10 | 0.90 | 18.00 | 2.00 | 1.97 | 1.11 → 0.84 |
| FC-V | 27 | - | - | - | - | - | - | - | - | - | - | - | - | 1.50 | 0.50 | - | 0.10 | 0.90 | 18.00 | 4.00 | 3.94 | 1.15 → 0.47 |

$^a$Limiting current density $i_{lim}$ represents the maximum current density that could potentially be reached given a specific reactant supply. $i_{lim} = 2FP_O X_{in}^{H_2O/H_2} V_{in}^{fuel} H_{ch} W_{ch}/(L_{cell} W_{cell} RT_{in})$.

$^b$This indicates the power variations from the initial state to the final steady state. Negative values denote the consumed power, and positive values denote the generated power.

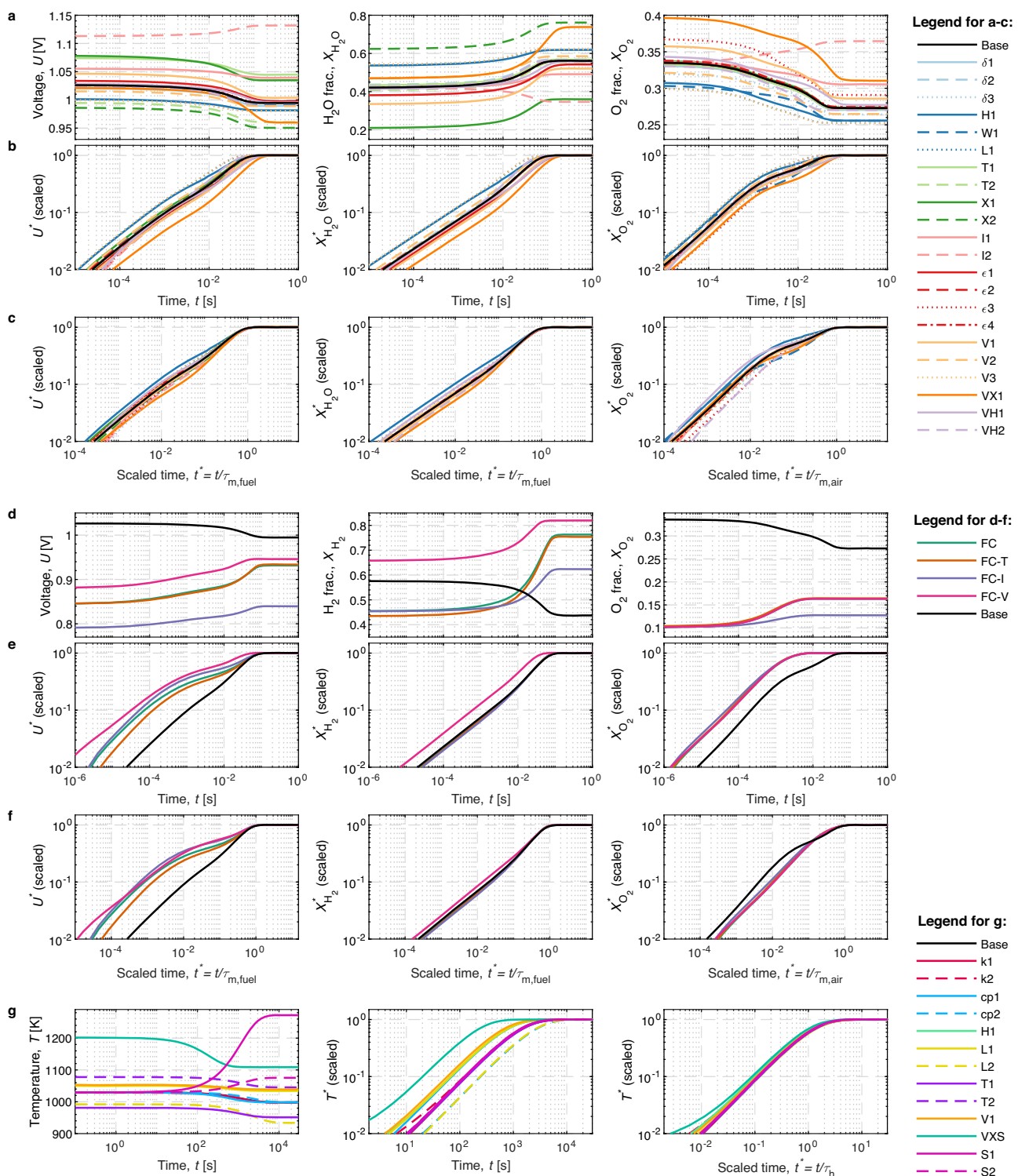

**Fig. 2 | Transient responses of SOCs. a–f** show the responses after step changes of current (mass source) at 0s. SOCs with different dimensions (*i.e.*, thickness -- $\delta$, height -- $H$, width -- $W$, length -- $L$), operating conditions (*i.e.*, temperature -- $T$, inlet species -- $X$, current -- $I$, inlet velocity -- $V$), and material properties (*i.e.*, porosity -- $\varepsilon$) are compared. Panels **a–c** present cases under electrolysis mode, while panels **d–f** present the fuel-cell (FC) mode operations. **a, d** The responses of voltage and the mole fractions of steam, hydrogen, and oxygen in functional layers. **b, e** The responses of voltage and mole factions are scaled according to: $U^* = (U - U_{t\to 0^+})/(U_{t\to\infty} - U_{t\to 0^+})$,

$X^*_{H_2O/H_2/O_2} = (X_{H_2O/H_2/O_2} - X_{H_2O/H_2/O_2,t=0})/(X_{H_2O/H_2/O_2,t\to\infty} - X_{H_2O/H_2/O_2,t=0})$, **c, f** Both the response variables and time are scaled. **g** shows the responses after step changes of heat source at 0 s. Temperature responses of cells with different material properties (*i.e.*, thermal conductivity -- $k$, specific heat capacity -- $c_p$), dimensions (*i.e.*, height -- $H$, length -- $L$), and operating conditions (*i.e.*, temperature -- $T$, inlet velocity -- $V$, inlet species -- $X$, heat source -- $S$) are compared. The scaled temperature is calculated by: $T^* = (T - T_{t=0})/(T_{t\to\infty} - T_{t=0})$. The case names and their specifics can be found in Tables 1 and 2.

**Table 2 | Manipulated variables used in the parametric study of heat transfer ('-' indicates that the parameter is consistent with the base case)**

| Name | Case* | $k_{DL}^{fuel}$ [W m⁻¹ K⁻¹] | $c_{p,int}$ [J kg⁻¹ K⁻¹] | $H_{ch}$ [m] | $L_{cell}$ [m] | $T_{in}$ [K] | $V_{in}^{fuel}$ [m s⁻¹] | $V_{in}^{air}$ [m s⁻¹] | $X_{in}^{H_2O}$ | $X_{in}^{H_2}$ | $S_{H,t=0}^{total}$ [W] | $S_{H,t>0}^{total}$ [W] |
|---|---|---|---|---|---|---|---|---|---|---|---|---|
| Base | 1 | 6 | 550 | 1.0E-03 | 0.10 | 1073.15 | 2.00 | 2.00 | 0.70 | 0.30 | -0.0792 | -0.135 |
| k1 | 2 | 0.6 | – | – | – | – | – | – | – | – | – | – |
| k2 | 3 | 60 | – | – | – | – | – | – | – | – | – | – |
| cp1 | 4 | – | 225 | – | – | – | – | – | – | – | – | – |
| cp2 | 5 | – | 1100 | – | – | – | – | – | – | – | – | – |
| H1 | 6 | – | – | 2.0E-03 | – | – | – | – | – | – | – | – |
| L1 | 7 | – | – | – | 0.05 | – | – | – | – | – | – | – |
| L2 | 8 | – | – | – | 0.20 | – | – | – | – | – | – | – |
| T1 | 9 | – | – | – | – | 1073.15 | – | – | – | – | – | – |
| T2 | 10 | – | – | – | – | 1123.15 | – | – | – | – | – | – |
| V1 | 11 | – | – | – | – | – | 4.00 | – | – | – | – | – |
| VXS | 12 | – | – | – | – | – | – | 18.00 | 0.10 | 0.90 | 1.352 | 0.370 |
| S1 | 13 | – | – | – | – | – | – | – | – | – | – | 0.370 |
| S2 | 14 | – | – | – | – | – | – | – | – | – | – | 0 |

*Case 1 to 11 exemplify endothermic operations of SOEC; Case 12 illustrates an exothermic operation of SOFC; Case 13 illustrates a transition from endothermic to exothermic operations. Case 14 represents a scenario where SOC switches from endothermic to thermal neutral, the Joule heat source in the electrolyte closely matches the heat sink in FLs for $t > 0$.

- The total heat source $S_H^{total}$ is assumed to be concentrated on AFL, CFL, and electrolyte, thereby neglecting the heat sources from other components.

A parametric study was conducted by changing the dimensions, thermal properties, and operating conditions of SOEC and SOFC, as presented in Table 2. Each simulation case is initialized with its steady state solution, where the total heat source $S_{H,t=0}^{total}$ is applied in AFL, CFL, and electrolyte. The heat source is then changed to $S_{H,t>0}^{total}$ in a small time step ($< 0.01$ s) and held constant until the end of simulation. The average temperature of FL in the SOEC or SOFC with respect to time are presented in Fig. 2g. The dimensionless temperature $T^*$, defined in the caption of Fig. 2, represents the extent to which the new steady state has been achieved, e.g., $T^* = 1$ indicates the temperature has reached the new steady state. As shown in Fig. 2g, the relaxation time of SOC after a step change in heat source varies widely from $10^2$ s to $10^3$ s.

To elucidate the relationship between thermal transients and various parameters in SOCs, the control volume concept with energy conservation principle is applied (see Supplementary Methods for detailed derivation). Given the boundary conditions of the system under investigation, there are only two inlets and two outlets that exchange heat with the environment, while the other boundaries are adiabatic. Thus, the total inflow heat transfer rate of the system can be represented by the sum of the enthalpy of fluids at the two inlets. Subsequently, the characteristic time of heat transfer within SOCs, denoted as $\tau_h$, was derived as follows,

$$\tau_h = \frac{\text{Total enthalpy} \quad [J]}{\text{Total heat transfer rate} [W]} = \frac{\mathcal{H}_0}{\dot{\mathcal{H}}_0}$$

$$= \frac{\mathcal{H}_{0,int}^{solid} + \mathcal{H}_{0,E}^{solid} + \mathcal{H}_{0,AFL}^{eff} + \mathcal{H}_{0,ADL}^{eff} + \mathcal{H}_{0,CDL}^{eff} + \mathcal{H}_{0,CFL}^{eff} + \mathcal{H}_{0,ch}^{fluid}}{\dot{\mathcal{H}}_{in}^{fuel} + \dot{\mathcal{H}}_{in}^{air}}$$

$$\approx \frac{(mc_p)_{0,int}^{solid} + (mc_p)_{0,E}^{solid} + (mc_p)_{0,AFL}^{eff} + (mc_p)_{0,ADL}^{eff} + (mc_p)_{0,CDL}^{eff} + (mc_p)_{0,CFL}^{eff} + (mc_p)_{0,ch}^{fluid}}{(\dot{m}c_p)_{in}^{fuel} + (\dot{m}c_p)_{in}^{air}}$$

(2)

where the superscript 'eff' indicates the properties of porous media that are calculated by volumetrically averaging the fluid and solid properties, the subscript '0' indicates the values are calculated based on the inlet temperature, $\dot{\mathcal{H}}$ denotes the enthalpy flow rate of fluid, $m$

denotes mass, and $\dot{m}$ denotes mass flow rate. The physical interpretation of $\tau_h$ is the time required to 'fill' the total enthalpy of the system via the inflow heat transfer rate.

When the time is scaled with $\tau_h$, the $T^*$–$t^*$ curves converge, as illustrated in Fig. 2g. This convergence indicates a comparable relaxation time of approximately $2\tau_h$ of all cases, underscoring the effectiveness of $\tau_h$ as a representative of the overall heat-transfer characteristic time in both SOEC and SOFC systems.

**Validation of the proposed characteristic times**

The proposed characteristic times for heat and mass transfer are validated using data from our experiments, as well as from existing literature (refer to Supplementary Results for details). Equations (1) and (2) are utilized to compute the characteristic times $\tau_m$ and $\tau_h$ for the SOC used in our experiments and for those discussed in other studies. The calculated values for $\tau_m$ and $\tau_h$ were then compared with the relaxation times that were either measured in our experiments or reported in literature, as tabulated in Table 3. The calculated characteristic times $\tau_m$ and $\tau_h$ were found to be of the same order of magnitude as the measured or reported relaxation times. This consistency was maintained across different application scenarios, such as planar and tubular SOCs, co-electrolysis, and even the PEMFC, a type of low-temperature electrochemical cell. These results not only further affirm the general applicability of the proposed characteristic times in representing the dynamic gaseous and thermal responses of various SOCs, but also indicate a potential for characterizing the dynamic processes of electrochemical cells with similar cell structures and flow phenomena to SOCs, such as PEMFCs and protonic ceramic fuel cells/electrolyzers.

**Application scenario: guidance of design adjustment of SOC**

The design of SOC involves a multitude of parameters. The expressions of $\tau_h$ and $\tau_m$, Eqs. (1) and (2), describe the relative importance of parameters. Taking the base case of SOEC (Table 1) as an example, Fig. 3a compares the void volume fraction of the components of SOC. Due to their lower contributions to the void volume, the microstructure and dimensions of DLs and FLs have a less significant impact on $\tau_m$ when compared to the dimensions of the fluid channel. As the numerator in Eq. (1), the volumetric flow rate significantly affects $\tau_m$. These observations align with the findings reported by Bae et al.[25]. In addition, Fig. 3a

**Table 3 | Comparison of calculated characteristic times and reported relaxation times for various cells with diverse operating conditions**

| Cell type | Response variables | Reported relaxation time [s] | Calculated $\tau_m$, $\tau_h$ [s][+] |
|---|---|---|---|
| Planar SOFC (in-house experiment)[*] | Voltage | $\approx \begin{cases} 9.33, & 100 \text{ sccm } H_2, 100 \text{ sccm } N_2 \\ 18.78, & 100 \text{ sccm } H_2 \\ 9.64, & 200 \text{ sccm } H_2 \\ 5.08, & 400 \text{ sccm } H_2 \end{cases}$ | $\tau_m^{fuel} = \begin{cases} 6.22 \\ 12.45 \\ 6.22 \\ 3.11 \end{cases}$ |
| 3-D planar SOFC[30] | Current | $\approx 0.6^a$ | $\tau_m^{fuel} = 0.46$ |
| 2-D tubular SOEC[16] (co-electrolysis) | $H_2O$ concentration | $0.255 - 1^b$ | $\tau_m^{fuel} = 0.42$ |
| | Temperature | $515 - 2000^b$ | $\tau_h = 2076$ |
| 3-D planar SOFC[19] | Fuel utilization | $\approx 0.7^a$ | $\tau_m^{fuel} = 0.49$ |
| | Air utilization | $\approx 0.05^a$ | $\tau_m^{air} = 0.04$ |
| Quasi 2-D SOEC stack[29] (co-electrolysis, 40 cells) | Current | $< 1^b$ | $\tau_m^{fuel} = 0.23$ |
| | Stack temperature | $> 900^b$ | $\tau_h = 3100$ |
| Flat-tube SOC[44] ($CO_2$ electrolysis) | Electrical impedance | $\approx 1.0^c$ | $\tau_m^{fuel} = 0.97$ |
| Planar PEMFC[45] (experiment) | Current | $\approx 1.0^d$ | $\tau_m^{fuel} = 0.41$ |

[+]Detailed calculation procedures are provided in Supplementary Tables 1 and 2.

[*]We conducted an in-house experiment in which the fuel was supplied to a 4 cm × 4 cm cell at different rates, denoted as standard cubic centimeters per minute (sccm). Details of the experiment are provided in Supplementary Results.

[a]The relaxation time is interpreted from figures.

[b]The relaxation time is interpreted from text and figures.

[c]The relaxation time is acquired from the first peak (lowest frequency) in the Distribution of Relaxation Time (DRT) figure.

[d]Cho et al.[45] noticed two time delays from the voltage response following a change of current. The first time delay is on the order of 1s.

presents a comparison of the heat capacities of various components in the base-case SOC. Of all the components, the interconnect has the largest heat capacity and thus plays a dominant role in determining the value of $\tau_h$. Moreover, the fluid flow rates, as the numerator in Eq. (2), also critically influence the thermal response time. This observation aligns with the findings of Liu et al.'s study[20] on thermal control.

Using the characteristic times $\tau_h$ and $\tau_m$ as guides, we modified the dimensions, inlet flow rates, and material properties of the base-case SOC. For the adjusted SOEC, we halved $H_{ch}$ and doubled $V_{in}^{fuel}$ to reduce the void volume while maintaining the overall volumetric flow rate and fuel utilization. These modifications resulted in a reduced value for $\tau_m^{fuel}$ and consequently, a faster gaseous response in the adjusted SOEC. Furthermore, we increased $V_{in}^{air}$ tenfold and halved $c_{p,int}$ in the adjusted SOEC to decrease $\tau_h$, which in turn expedited the thermal response.

Figure 3 b,c compare the responses between the base and adjusted cases. When the current magnitude $|i|$ underwent a step change from 1 A cm$^{-2}$ to 0.5 A cm$^{-2}$, both cases exhibit delayed responses of $X_{H_2O}$ and $T$ due to heat and mass transfer lags[15,34]. However, the adjusted SOEC showed significantly faster electrical, gaseous, and thermal responses. The prediction based on $\tau_m^{fuel}$ indicates that the relaxation time of $X_{H_2O}$ will decrease from 0.088s to 0.063s after design adjustment. This prediction is consistent with the simulation result presented in Fig. 3c, where the relaxation time of $X_{H_2O}$ decreases from 0.091s to 0.063s. This agreement indicates that the proposed characteristic times can serve as precise and quantifiable indicators in SOC design. In terms of thermal responses, the relaxation time of $T$ is predicted to decrease from 2187 s to 370 s, compared to the simulation result that decreases from 1726 s to 438 s. This fact implies that $\tau_h$ can only provide a rough estimate of the thermal relaxation time following a step change of current. This is due to the complexity introduced by the varying heat source of SOEC in response to dynamic electrical conditions.

**Application scenario: guidance of SOC control**

In our previous study[15], we found that heat and mass transfer lags can cause current undershoot after rapid voltage ramps on SOEC. We suggested reducing the voltage ramp time to mitigate current undershoot induced by mass-transfer lag, as shown in Fig. 3d. However, the suggested ramp time would vary for different SOCs and cannot be determined without conducting experiments or simulations. This limitation impairs the generalizability of the control strategy for SOC. Our current study provides a convenient method for estimating the ramp time required to alleviate the undershoot by using Eq. (1). The ramp time is recommended to be $1\tau_m^{fuel}$, equivalent to the relaxation time of $X_{H_2O}$. As shown in Fig. 3d, slowing down the voltage ramp time to be $1\tau_m^{fuel}$ can reduce most of the current undershoot induced by the mass-transfer lag. While further decreasing the ramp time will have minor effects in mitigating the undershoot. However, it is difficult to reduce the undershoot induced by heat-transfer lag by simply slowing down the voltage ramp due to the large value of $\tau_h$. In conclusion, the proposed characteristic times in this study enhances the generalizability of control strategies by offering a quantifiable metric that can be applied uniformly across various SOCs.

## Discussion

In summary, we proposed general expressions of two characteristic times ($\tau_h$ and $\tau_m$) to represent the overall heat and mass transfer rates within various SOCs under different operating conditions. The gaseous and thermal response times upon rapid electrical variations are at the order of $\tau_m$ and $\tau_h$, respectively. The effectiveness of $\tau_h$ and $\tau_m$ was also validated against our numerical simulations and experiments, as well as the transient characteristics of several types of SOCs and a PEMFC reported in the literature. In terms of potential applications, on one hand, $\tau_h$ and $\tau_m$ can serve as quantifiable and easy-to-calculate indicators for designing SOCs with desired transient characteristics. On the other hand, $\tau_m$ can enhance the generalizability of existing SOC control strategies. The proposed characteristic times could enrich the theoretical comprehension of SOCs, especially during unstable operating conditions, and could potentially boost the development of SOC for renewable energy storage and conversion.

In terms of the methodology to derive the characteristic times, we adopted the non-dimensional analysis to identify the parameters and time constants that directly influence SOC transients. This methodology leads to an innovative theoretical framework to study the heat and mass transfer phenomena in SOCs, which may also spark new insights

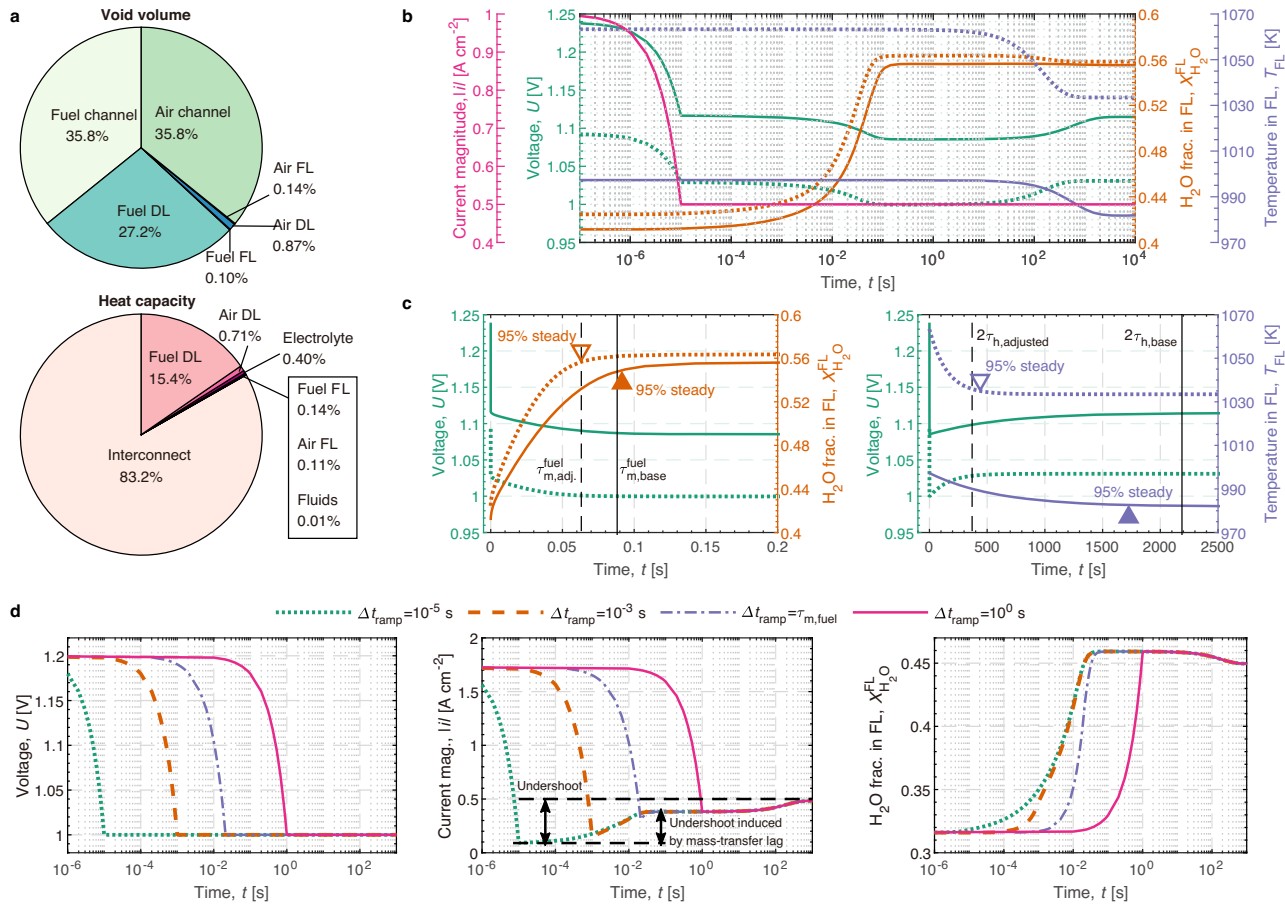

**Fig. 3 | Guidance of SOC design and control using the proposed characteristic times. a** Relative importance of different components for heat and mass transfer rates. **b** Comparison on the transient responses of two SOECs after the step changes of current in a logarithmic timeline. **c** The responses in linear timeline. The solid line represents the base case of SOEC, and the dashed line represents the adjusted SOEC. **d** Responses of SOEC to voltage ramps with different ramp rates[15].

into the transient behavior of electrochemical cells with similar structures and flow phenomena, such as proton exchange membrane fuel cells/electrolyzers and protonic ceramic fuel cells/electrolyzers.

In addition, the proposed characteristic times are intended to generalize the response times in various SOCs. To acquire the exact values of response times, the local effects of SOC designs, such as the cross-sectional shape of channel[35], should be considered. For future work, it would be intriguing to compare the transient performance of different channel designs that share the same general characteristic times. In such research, the generalized characteristic time could serve as a benchmark to facilitate channel optimization.

## Methods

To fulfill the research objectives, we develop a comprehensive five-step analysis procedure, as depicted in Fig. 1b. This section provides a brief introduction to the numerical model employed in our study, and the non-dimensional analysis carried out on the fundamental governing equations to identify key parameters influencing the transient behaviors of SOCs. Next, a parametric study is conducted using the three-dimensional (3-D) transient numerical model to quantify the effect of each key parameter. In the Results Section and Supplementary Methods, we derive generalized characteristic times for heat and mass transfer in SOC with clear physical interpretations.

### Numerical modeling

The 3-D Computational Fluid Dynamics (CFD) model, as shown in Fig. 1a, has been previously developed and validated in our work[15]

using Ansys Fluent R18.1. The adoption of a 3-D single-channel model reduces computational cost while maintaining high spatial resolution[25,36].

In our model[15], Fick's law is utilized to model gas diffusion within fluid channels, a process inherently influenced by variations in temperature and pressure. In porous media, the realistic microstructures of electrodes are neglected, and the microstructural and Knudsen effects on mass diffusion are considered through the extended Fick's law[33]. Besides, the convection of momentum, mass, and heat is also considered in porous media. Electrochemical reactions are modeled by employing the Butler-Volmer equation, which is derived from single-step and single-electron-transfer reactions and shows sufficient accuracy when calculating the multi-electron and multi-step reactions in SOCs[15,33]. Fluid densities are computed using the ideal gas law. The temperature-dependent fluid properties (e.g., viscosity, specific heat, and thermal conductivity) at ambient pressure are sourced from the NIST database[37] and integrated into the model *via* piecewise linear interpolation. In terms of the solid materials, the ionic, electronic, and thermal properties are constant, except for the temperature-dependent ionic conductivity of the electrolyte. The effective properties of porous media are determined by averaging the fluid and solid properties volumetrically.

A set of conservation equations for momentum, species mass, energy, electronic charge, and ionic charge are coupled and solved to obtain transient solutions for velocity, species mass fractions, temperature, and electrical and ionic potentials within the SOC. In addition, an adaptive time-stepping algorithm is employed to guarantee adequate

temporal resolution for capturing electrical behaviors during the transient simulation. The time-step size at each step is constrained by the variation rates of species concentration and temperature. Further details of the numerical model can be found in our previous study[15].

**Identifying key parameters by non-dimensional analysis**

Non-dimensional analysis is a widely used technique in Fluid Mechanics that enables the generalization of crucial information about similar physical phenomena[31]. Several studies[38–41] have employed non-dimensional analysis to examine one-dimensional (1-D) and two-dimensional (2-D) models of SOC, leading to a deeper understanding of steady-state transport phenomena within SOCs. To study the transient characteristics, Achenbach[26] explored the relationship between the electrical responses of SOFCs and the dimensionless time scale of heat transfer through a 3-D model, but the transients of mass transfer were not investigated. Therefore, in this work, non-dimensional analysis is applied to a 3-D SOC model to investigate transient heat transfer and mass transfer processes, aiming to identify the global time constants and the corresponding key parameters that directly influence SOC transients.

To simplify the non-dimensional analysis of the governing equations, we have made the following two assumptions. Firstly, fluid properties are assumed to remain constant in dimensional analysis, allowing properties such as density $\rho$, diffusivity $D$, thermal

conductivity $k$, and specific heat $c_p$ to be taken out of the partial differential terms. Secondly, the thermal transport term due to mass diffusion is omitted in the energy equations since its contribution is small. The simplified governing equations, scales for all variables, and the non-dimensional equations are listed in Table 4. To non-dimensionalize time, we introduce two characteristic times $\tau_m$ and $\tau_h$ to scale the response times of mass and heat, respectively. The mathematical expressions for $\tau_m$ and $\tau_h$ are proposed in the Results Section. For other variables, the scaling rules are as follow:

- The variation of the mass fraction $Y_i$, where $i$ denotes the species of interest, is scaled by the difference between its initial mass fraction at $t = 0$ and its final steady state mass fraction at $t \to \infty$. This scaling rule is also applied to obtain the dimensionless mole fraction $X_i^*$, and dimensionless temperature $T^*$.
- Spatial dimensions are scaled by the corresponding dimensions of the cell: $y$ is scaled by the length of the cell $L_{cell}$, $x$ by the width of the cell $W_{cell}$, $z$ by the thickness or height of the corresponding domain.
- The inlets of anode and cathode fluid channels are selected as the reference states. Velocity $V$ is scaled by the inlet velocity $V_{in}$.

According to the dimensionless equations presented in Table 4, the variations of species mass fraction $Y$ and temperature $T$, which

## Table 4 | Non-dimensional analysis of the heat and mass transfer processes in SOC

| Simplified governing equations for non-dimensional analysis[15,40] | Domains |
|---|---|
| *Conservation of species* | |
| $\frac{\partial Y_i}{\partial t} + \vec{V} \cdot \nabla Y_i - D\nabla^2 Y_i = 0$ | Fluid channel |
| $\frac{\partial(\varepsilon Y_i)}{\partial t} + \vec{V} \cdot \nabla Y_i - D_{eff}\nabla^2 Y_i = 0$ | ADL,CDL |
| $\frac{\partial(\varepsilon Y_i)}{\partial t} + \vec{V} \cdot \nabla Y_i - D_{eff}\nabla^2 Y_i = \frac{S_{m,i}}{\rho}$ | AFL,CFL |
| *Conservation of energy* | |
| $\frac{\partial T}{\partial t} + \vec{V} \cdot \nabla T - \alpha\nabla^2 T = 0$ | Fluid channel |
| $\frac{\partial T}{\partial t} + \frac{\rho c_p}{\rho_{eff} c_{p,eff}} \vec{V} \cdot \nabla T - \alpha_{eff}\nabla^2 T = \frac{S_h}{\rho_{eff} c_{p,eff}}$ | Porous[a] |
| $\frac{\partial T}{\partial t} - \alpha_s\nabla^2 T = \frac{S_{h,s}}{\rho_s c_{p,s}}$ | Solid[b] |
| **Scaling & definitions[c]** | |
| $t^* = \frac{t}{\tau_m}$ or $\frac{t}{\tau_h}$, $Y_i^* = \frac{Y_i - Y_{i,t=0}}{\Delta Y_i} = \frac{Y_i - Y_{i,t=0}}{Y_{i,t\to\infty} - Y_{i,t=0}}$, $X_i^* = \frac{X_i - X_{i,t=0}}{X_{i,t\to\infty} - X_{i,t=0}}$, $V^* = \frac{V}{V_{in}}$, $T^* = \frac{T - T_{t=0}}{\Delta T} = \frac{T - T_{t=0}}{T_{t\to\infty} - T_{t=0}}$, $\nabla^* = L_{cell}\nabla$, $x^* = \frac{x}{W_{cell}}$, | |
| $y^* = \frac{y}{L_{cell}}$, $z^* = \frac{z}{H_{ch}}$ or $\frac{z}{\delta_{DL}}$ or $\frac{z}{\delta_{FL}}$, $S_{m,i}^* = \frac{S_{m,i}\tau_m}{\rho_0 \Delta Y_i}$, $S_h^* = \frac{S_h\tau_h}{\rho_{eff} c_{p,eff}\Delta T}$ or $\frac{S_h\tau_h}{\rho_s c_{p,s}\Delta T}$, $\alpha = \frac{k}{\rho c_p}$, $\alpha_{eff} = \frac{k_{eff}}{\rho_{eff} c_{p,eff}}$, $\alpha_s = \frac{k_s}{\rho_s c_{p,s}}$ | |
| **Dimensionless equations** | **Domains** |
| *Conservation of species* | |
| $\frac{\partial Y_i^*}{\partial t^*} + \frac{\tau_m}{L_{cell}/V_{in}} \vec{V}^* \cdot \nabla^* Y_i^* - \left( \frac{\tau_m}{W_{cell}^2/D} \frac{\partial^2 Y_i^*}{\partial x^{*2}} + \frac{\tau_m}{L_{cell}^2/D} \frac{\partial^2 Y_i^*}{\partial y^{*2}} + \frac{\tau_m}{H_{ch}^2/D} \frac{\partial^2 Y_i^*}{\partial z^{*2}} \right) = 0$ | Fluid channel |
| $\varepsilon\frac{\partial Y_i^*}{\partial t^*} + \frac{\tau_m}{L_{cell}/V_{in}} \vec{V}^* \cdot \nabla^* Y_i^* - \left( \frac{\tau_m}{W_{cell}^2/D_{eff}} \frac{\partial^2 Y_i^*}{\partial x^{*2}} + \frac{\tau_m}{L_{cell}^2/D_{eff}} \frac{\partial^2 Y_i^*}{\partial y^{*2}} + \frac{\tau_m}{\delta_{DL}^2/D_{eff}} \frac{\partial^2 Y_i^*}{\partial z^{*2}} \right) = 0$ | ADL,CDL |
| $\varepsilon\frac{\partial Y_i^*}{\partial t^*} + \frac{\tau_m}{L_{cell}/V_{in}} \vec{V}^* \cdot \nabla^* Y_i^* - \left( \frac{\tau_m}{W_{cell}^2/D_{eff}} \frac{\partial^2 Y_i^*}{\partial x^{*2}} + \frac{\tau_m}{L_{cell}^2/D_{eff}} \frac{\partial^2 Y_i^*}{\partial y^{*2}} + \frac{\tau_m}{\delta_{FL}^2/D_{eff}} \frac{\partial^2 Y_i^*}{\partial z^{*2}} \right) = S_{m,i}^*$ | AFL,CFL |
| *Conservation of energy* | |
| $\frac{\partial T^*}{\partial t^*} + \frac{\tau_h}{L_{cell}/V_{in}} \vec{V}^* \cdot \nabla^* T^* - \left( \frac{\tau_h}{W_{cell}^2/\alpha} \frac{\partial^2 T^*}{\partial x^{*2}} + \frac{\tau_h}{L_{cell}^2/\alpha} \frac{\partial^2 T^*}{\partial y^{*2}} + \frac{\tau_h}{H_{ch}^2/\alpha} \frac{\partial^2 T^*}{\partial z^{*2}} \right) = 0$ | Fluid channel |
| $\frac{\partial T^*}{\partial t^*} + \frac{\tau_h}{L_{cell}/V_{in}} \frac{\rho c_p}{\rho_{eff} c_{p,eff}} \vec{V}^* \cdot \nabla^* T^* - \left( \frac{\tau_h}{W_{cell}^2/\alpha_{eff}} \frac{\partial^2 T^*}{\partial x^{*2}} + \frac{\tau_h}{L_{cell}^2/\alpha_{eff}} \frac{\partial^2 T^*}{\partial y^{*2}} + \frac{\tau_h}{\delta_{DL}^2/\alpha_{eff}} \frac{\partial^2 T^*}{\partial z^{*2}} \right) = S_h^*$ | ADL, CDL |
| $\frac{\partial T^*}{\partial t^*} + \frac{\tau_h}{L_{cell}/V_{in}} \frac{\rho c_p}{\rho_{eff} c_{p,eff}} \vec{V}^* \cdot \nabla^* T^* - \left( \frac{\tau_h}{W_{cell}^2/\alpha_{eff}} \frac{\partial^2 T^*}{\partial x^{*2}} + \frac{\tau_h}{L_{cell}^2/\alpha_{eff}} \frac{\partial^2 T^*}{\partial y^{*2}} + \frac{\tau_h}{\delta_{FL}^2/\alpha_{eff}} \frac{\partial^2 T^*}{\partial z^{*2}} \right) = S_h^*$ | AFL, CFL |
| $\frac{\partial T^*}{\partial t^*} - \left( \frac{\tau_h}{W_{cell}^2/\alpha_s} \frac{\partial^2 T^*}{\partial x^{*2}} + \frac{\tau_h}{L_{cell}^2/\alpha_s} \frac{\partial^2 T^*}{\partial y^{*2}} + \frac{\tau_h}{\delta_s^2/\alpha_s} \frac{\partial^2 T^*}{\partial z^{*2}} \right) = S_h^*$ | Solid[b] |

[a]Porous media includes ADL, AFL, CDL, and CFL.
[b]Solid domain includes interconnect and solid oxide electrolyte.
[c]The subscript 'eff' denotes the effective properties of porous media, which are calculated by averaging the fluid and solid properties volumetrically[15]. For example, the effective thermal conductivity of porous media is calculated as $k_{eff} = \varepsilon k + (1-\varepsilon)k_s$.

**Table 5 | Boundary conditions for the numerical simulation (the operating pressure is $p_O$ = 1 atm)**

|  | Inlet[a] | Outlet | $x = 0$, $x/W_{cell} = 1$ | $z = 0$ | $z/\delta_{cell} = 1$ | Other surfaces |
|---|---|---|---|---|---|---|
| Momentum | $V_{in}^{air}$, $V_{in}^{fuel}$ | $p_{gauge} = 0$ | Zero flux | N.A. | N.A. | $V = 0$ |
| Thermal | $T_{in}$ | N.A. | Zero flux | Zero flux | Zero flux | Zero flux |
| Species | $X_{in}^{O_2} = 0.2$, $X_{in}^{N_2} = 0.8$, $X_{in}^{H_2O}$, $X_{in}^{H_2}$ | N.A. | Zero flux | N.A. | N.A. | Zero flux |
| Electrical | N.A.* | N.A. | Zero flux | $\phi_{ele} = 0$ | Specified flux[b] | Zero flux |

[a]The cell encompasses two inlets: the fuel channel inlet and the air channel inlet. The fuel inlet solely comprises $H_2O$ and $H_2$, while the air inlet solely comprises $O_2$ and $N_2$. In the simulation, the two inlets will have different velocities but same temperature.

[b]The electric current density $i$ on the surface is a time-dependent function. $i$ initiates at $i_{t=0}$ and transitions to $i_{t>0}$ in a very short time, and then maintains at $i_{t>0}$.

*N.A. means 'not applicable'. $V_{in}$, $T_{in}$, $X_{H_2O}$, $X_{H_2}$, $i_{t=0}$, and $i_{t>0}$ are manipulated variables in the parametric study, while the other boundary conditions are unchanged.

reflect the mass and heat transfer in SOC, are influenced by the following parameters: (1) cell dimensions (i.e., $W_{cell}$, $L_{cell}$, $H_{ch}$, $\delta_{ADL}$, $\delta_{AFL}$, $\delta_{CDL}$, and $\delta_{CFL}$); (2) operating conditions including flow conditions (i.e., $V_{in}$, $T_{in}$, and $X_{H_2O/O_2/H_2,in}$) and electrical conditions (i.e., current $i$ and voltage $U$) that determine the heat and mass sources; (3) fluid properties (i.e., $D$, $\rho$, $c_p$, and $\alpha$) and material properties (i.e., $\rho_s$, $c_{p,s}$, $k_s$, $\alpha_s$, $\varepsilon_{ADL}$, $\varepsilon_{AFL}$, $\varepsilon_{CDL}$, and $\varepsilon_{CFL}$).

Besides, in Table 4, we identified the time ratios (bold terms) that govern the transient solutions of $Y_i^*$ and $T^*$. Each time ratio has a denominator representing the time scale of a local transport phenomenon in a specific location[42]. For example, $\delta_{DL}^2/D_{eff}$ represents the gas diffusion characteristic time across the thickness of DL, $H_{ch}^2/\alpha$ represents the thermal diffusion characteristic time across the height of channel, and $L_{cell}/V_{in}$ is the convection characteristic time. In addition, $\tau_m$ or $\tau_h$ serves as the numerator of all time ratios, representing the global characteristic time of mass/heat transfer within the entire SOC. Therefore, the expressions of $\tau_m$ and $\tau_h$ are critical for generalizing SOC transient characteristics but their expressions remain unknown in non-dimensional analysis. In this study, their generalized expressions are derived based on the results of parametric study as presented in Results Section.

### Parametric study of the dynamic responses of SOCs

To comprehensively analyze the impact of various parameters on the transient characteristics of SOC, a meticulous parametric study was conducted through numerical simulations by altering the dimensions, material properties, and operating conditions of SOC. The boundary conditions (BC) of all simulations are presented in Table 5. The parametric study consists of two parts, one focusing on the transient mass-transfer behavior (see Table 1) and the other on the thermal behavior (see Table 2). The results of the parametric study are presented in the Results Section.

### Reporting summary

Further information on research design is available in the Nature Portfolio Reporting Summary linked to this article.

## Data availability

The data that support the plots within this paper are available in the Source data file. Additional data are available from authors upon request. Source Data file has been deposited in Figshare under accession code DOI link[43]. Source data are provided with this paper.

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

## Acknowledgements

The authors gratefully acknowledge the partial support from The Hong Kong Polytechnic University (P0035016); National Natural Science Foundation of China (No. 52306236); Science, Technology and Innovation Commission of Shenzhen Municipality (GXWD20220811165757005); and Department of Education of Guangdong Province (2021KQNCX271). The authors acknowledge GPT-4 and New Bing for their assistance in improving the language quality of this paper.

## Author contributions

Z.L.: Conceptualization, Formal Analysis, Investigation, Methodology, Software, Writing - Original Draft. J.W.: Funding Acquisition, Writing - Review & Editing. K.R.: Experiment, Writing - Review & Editing. Z.J.: Experiment, Writing - Review & Editing. M.N.: Investigation, Writing - Review & Editing. L.A.: Supervision, Writing - Review & Editing. Y.W.: Software. J.Y.: Writing - Review. M.L.: Funding Acquisition, Supervision, Conceptualization, Methodology, Writing - Review & Editing.

## Competing interests

The authors declare no competing interests.
