## [Peer Review File · Nature Communications]

REVIEWER COMMENTS

Reviewer #1 (Remarks to the Author):

The work presented is in deed of very high quality with the promised potential to actually revolutionize SOC development, optimization and control. Methodology is sound and definitely meet the standards in the field.

Work worthy of publication in nature communications however, after revision and implementation of proposed changes and discussion of critical aspects or solid and convincing rebutal.

One of such critical points that need to be discussed is the impact of decoupling SOEC heat sources from electrochemical reactions and not considering current density dependent joule heat located in the electrolyte and that actually ballances endothermic reaction heat at the thermoneutral point. This means there are infact two regions, OCV (current density = 0) and thermoneutral voltage (current density < 0) at which overall heat source value would be 0; At least voltage response at both points however cannot be same. This point should be clearly discussed in the article.

L5: Please start sentence with "Employment of a" ...

L6: Please start sentence with "Quantification of the relationship..."

L7: Please start sentence with "Identification of the characteristic times..."

L8: Please start sentence with "Validation of characteristic times against ..."

L9: Please start sentence with "Elucidation of the potential of identified characteristic times for SOC design and control optimization with minimal computational cost"

L36: please replace "store" with "convert" because SOEC "convert" electrical and thermal energies to chemical energy of produced hydrogen, carbon monoxide of syngas which can then be "stored". Thus SOECs only enable storage or convert

L45: "Nerat [19] recommended increasing the anode supporting layer thickness to 0.1 mm"

This is not correct!

Nerat finds that 0.1mm is prone to starvation. This means he recommends maintaining the anode support thickness above 0.1mm

This does not mean to increase the thickness to at least 0.1mm, because support layers of technical or state of the art anode supported cells are 0.3mm and above, down from 1.5mm, to 1mm, to 0.5mm to now 0.3mm over the years. To the best of my knowledge, there is no technically relevant Anode Supported Cell, with a support thickness lower than 0.1mm. Please reformulate the sentence.

L67: What do the authors mean with the phrase "the incorporation of advanced heat and mass transfer theory within SOCs may stimulate new comprehension". SOC literature is filled with heat and mass transfer theories. How advanced are these theories employed in this article

L68 – L70: I do not see the analogy in structure and flow phenomenon between group I consisting of SOC, PEMFC, PCEC, i.e. protonic ceramic fuel cell/electrolyzer and group II consisting of alkaline fuel cells and electrolyser and flow battery. Please either elaborate more or stick to group I systems.

L85: please replace "water" with "steam" because at the SOC operation temperatures, it is most definitely steam that is produced and not water

L89: please replace "voids" with "pores"

L90: please replace "exclusively" with "predominantly", because in SOCs with not so high performance, the penetration depth goes beyond the generally < 10 μm thick functional layers into the diffusion layers. This means the DLs have some degree of ionic conduction. This holds especially for the fuel electrodes, since some oxygen electrodes may have very high electronic conductive DL e.g. LSM but minimal ionic conductivity.

L91: please include "or double-phase" after "triple-phase" because in Ni/GDC fuel electrodes for instance, conversion also take place at the pore/GDC double phase boundary due to mixed conduction of GDC under reducing atmosphere

L187: Please replace “While both of the diffusive and convective time constants are inherently used to describe 1-D local transport phenomena. The time constants for 1-D transport ...” with

“While both diffusive and convective time constants are inherently used to describe 1-D local transport phenomena, the time constants for 1-D transport ...”

L197 and L208: the relaxation time of voltage is scaled with τ_m of fuel electrode and nicely shown that electrical response is limited by species transport in the fuel electrode. For sake of completeness, the authors should please include a scaling of the voltage response to τ_m of the air electrode for both SOEC and SOFC modes in corresponding figures. It should come out, that species transport in the air electrode is not limiting the electrical response. Otherwise, the authors should at least state that this was done and a value obtained (please state value) depicting the non-limitation of electrical response by species transport in the air electrode.

L212: Please remove “diffusivity of gas” and “material properties” from sentence. I doubt that “temperature exerts a profound impact” on diffusivity of gas and microstructure properties. Furthermore, from the cited reference, Reference 39, temperature impact on gas diffusivity in both electrodes is shown to be very small in both SOFC and SOEC between 750°C and 850°C compared to impact on activation overpotential

L216: please replace “vice versa” with “beyond it, as use of vice versa as such is rather unfortunate

L218: “To reduce this complexity, we decoupled the heat sources from the electrochemical model...”; The heat sources include the electrochemical reactions, but most importantly the joule heat especially arising from ionic flow across the electrolyte. This is the heat source that balances out the endothermicity to enable the thermoneutral point/voltage (ca. 1,29V) generally at very high current densities for good performance electrodes. How does employing constant values for heat source (cf. L223) solve the ambiguous situation in SOEC, in which the heat source value at OCV and thermoneutral point can be considered equal and having a value of zero. Is this a weakness of the concept, or does the presented method account for this problem, which seems quite important for the SOEC system design, operation, and control?

L224: Again, and stressing the point discussed above, the “majority of heat source or sink” is considered to come from the electrochemical reactions located in the functional layers. However, joule heat stems predominantly from the electrolyte and as mentioned actually equals and cancels out the electrochemical heat at the thermoneutral point at generally high current density. Should joule heat really be neglected or the proposed decoupling in deed account for it after all? How does this impact the electrical response around the thermoneutral voltage?

Table 5: Not clear which response variables belong to which Cell type, in the case of multiple response variables; Please either include table lines of use brackets or larger spacing for to distinguish Cell type

L260: Please replace “minor” with “lower”, because from figure 6, fuel electrode DL contributes 27.2% of void volume. Given that fuel and air channels contribute 35.8%, I would not call the contribution of at least the fuel electrode DL as “minor”. This can be employed for FLs or air electrode DL with percentages < 1;

Reviewer #2 (Remarks to the Author):

The methodology employed in this study is appropriately chosen, and the presented results are indeed remarkable. However, prior to considering the publication of this paper in this journal, I have outlined some comments to further enhance its quality:

1. It is suggested that in the concluding paragraph of the introduction section, a clear and comprehensive explanation of the novelty of your work and its primary objectives be provided.
2. A notable revision is required for the abstract to render it more specific and precise. Currently, it lacks the necessary level of detail required to effectively summarize the research
3. In the realm of mass transfer, it might be worth considering the incorporation of non-dimensional numbers such as the Sherwood number and Reynolds number. Similarly, for the heat transfer aspect, the inclusion of the Nusselt number could potentially offer deeper insights. Could you please elucidate the rationale behind the decision to not employ these non-dimensional numbers in your work?
4. It's of utmost importance to address whether the potential alteration of flow properties due to electrochemical reactions within the channel has been integrated into your model. This consideration holds the potential to impact the residence time for mass transport. A clarification regarding the inclusion of this phenomenon in your manuscript is necessary.
5. Given the utilization of a numerical method, it would be advantageous to visually represent the mesh independence through a graph or table based on non-dimensional parameters like the Sherwood number and Nusselt number.
6. Additionally, kindly provide information regarding the step size utilized in your numerical method.
7. A comprehensive description of all properties and formulations integrated into your numerical model is imperative. This encompasses aspects such as effective diffusion, membrane properties, and the like.
8. It is recommended to provide detailed explanations of membrane properties and other pertinent information, substantiated with appropriate references.

9.The omission of mentioning the limiting current in your manuscript is noted. Given its significance in the upscaling process, this parameter warrants inclusion and discussion.

10.The conclusion section necessitates revision in order to render your findings and their implications more transparent and concise.

By thoughtfully addressing these recommendations, your paper stands to significantly elevate its value and impact.

Ref.: NCOMMS-23-30425

General characteristic times of transient responses to changing voltage/current in solid oxide cells (SOCs): from theory to applications

We would like to thank the editor and all reviewers for their constructive comments, which helped us improve our manuscript. We address each comment individually below. Changes made according to editor and reviewers' comments are highlighted in the revised manuscript.

Response to Comments by Reviewer #1:

The work presented is in deed of very high quality with the promised potential to actually revolutionize SOC development, optimization and control. Methodology is sound and definitely meet the standards in the field.

Work worthy of publication in nature communications however, after revision and implementation of proposed changes and discussion of critical aspects or solid and convincing rebutal.

One of such critical points that need to be discussed is the impact of decoupling SOEC heat sources from electrochemical reactions and not considering current density dependent joule heat located in the electrolyte and that actually balances endothermic reaction heat at the thermoneutral point. This means there are infact two regions, OCV (current density = 0) and thermoneutral voltage (current density < 0) at which overall heat source value would be 0; At least voltage response at both points however cannot be same. This point should be clearly discussed in the article.

L218: "To reduce this complexity, we decoupled the heat sources from the electrochemical model..."; The heat sources include the electrochemical reactions, but most importantly the joule heat especially arising from ionic flow across the electrolyte. This is the heat source that balances out the endothermicity to enable the thermoneutral point/voltage (ca. 1,29V) generally at very high current densities for good performance electrodes. How does employing constant values for heat source (cf. L223) solve the ambiguous situation in SOEC, in which the heat source value at OCV and thermoneutral point can be considered equal and having a value of zero. Is this a weakness of the concept, or does the presented method account for this problem, which seems quite important for the SOEC system design, operation, and control

L224: Again, and stressing the point discussed above, the "majority of heat source or sink" is considered to come from the electrochemical reactions located in the functional layers. However, joule heat stems predominantly from the electrolyte and as mentioned actually equals and cancels out the electrochemical heat at the thermoneutral point at generally high current density. Should joule heat really be neglected or the proposed decoupling in deed account for it after all? How does this impact the electrical response around the thermoneutral voltage?

Thank you for these insightful comments. Regarding our model's simplifications, especially the heat source in the SOC thermal response, we made the following revisions:

1. We agree that the dominant heat sources/sinks are found within the Functional Layers (FLs) and the electrolyte. In our revised manuscript (Supplementary Results **Lines 127-135** & Supplementary Fig.2c), we have clarified this point:

“Notably, the dominant heat sources or sinks are found within the Functional Layers (FLs) and the electrolyte, while the contributions from other elements are relatively insignificant. Therefore, for convenience, our thermal investigation merely considers the heat sources in FLs and electrolyte.”

Supplementary Figure 2: ... c Comparison of heat sources in different elements at 1.1V and TNV.

2. We've made extensive revision to clarify our simulation strategy of the heat source:

Revised main text (Line 134)

“The total heat source S_H^{total} is assumed to be concentrated on AFL, CFL, and electrolyte...”

Supplementary Results (Lines 69-73)

“...AFL, electrolyte, and CFL are adjacent and exhibit minimal thicknesses, approximately 0.01 mm. Consequently, the temperatures across the AFL, electrolyte, and CFL are nearly uniform in the thickness direction. This uniformity allows for the thermal effects of AFL, CFL, and electrolyte to be approximated as a lumped heat source from a thin region. As such, we model the different thermal states of SOCs by adjusting the total heat source on FLs and electrolyte.”

Our analysis also implies that the distribution of heat source among AFL, electrolyte, and CFL is unimportant. To illustrate this, we have compared two scenarios with different heat source distributions but equal total heat source ($S_H^{\text{total}} = 0.3\text{W}$) on the AFL, electrolyte (E.), and CFL:

- Scenario A: $S_H^{\text{AFL}} = S_H^{\text{CFL}} = 0.15\text{ W}$, $S_H^{\text{E.}} = 0\text{ W}$
- Scenario B: $S_H^{\text{AFL}} = S_H^{\text{CFL}} = -0.1\text{W}$, $S_H^{\text{E.}} = 0.5\text{W}$

	Scenario A	Scenario B	Difference
	T_{ave} [K]	T_{ave} [K]	T [K]
Air interconnect	1235.1212	1235.1264	-0.0052
Air channel	1222.4062	1222.411	-0.0048
Air DL	1235.1955	1235.2006	-0.0051

Air FL	1235.2071	1235.2132	-0.0061
Electrolyte	1235.2074	1235.2166	-0.0092
Fuel FL	1235.207	1235.2145	-0.0075
Fuel DL	1235.1014	1235.1086	-0.0072
Fuel channel	1222.7678	1222.7745	-0.0067
Fuel interconnect	1234.9343	1234.9415	-0.0072

The results show that despite differences in the spatial distribution of heat sources, both scenarios result in nearly identical SOC temperature fields, supporting our initial simulation strategy of the heat source. Reference [2] holds the same view that the fraction of heat source assigned to each electrode has little effect on the final solution. In addition, the electrical responses of the two scenarios will be identical due to the almost identical temperature distribution.

Furthermore, in response to the reviewer's question, 'Should joule heat really be neglected...', we want to clarify that the Joule heat's thermal effect in the electrolyte S_H^E is not neglected. It is included in the total heat source S_H^{total} , one of our study's manipulated variables.

3. Regarding to the mentioned 'ambiguous situation' of OCV and TNV, we understand that these represent distinct electrical states but the same thermal state ($S_H^{\text{total}} = 0$). The reviewer might wonder which electrical state we are referring to when we apply $S_H^{\text{total}} = 0$ to SOCs, either the OCV or TNV. Actually, $S_H^{\text{total}} = 0$ refers to both the OCV and TNV situations. Our study focuses on the characteristic time of thermal response (τ_h) after the step change of heat source, not directly the step change of electrical conditions. The step change of heat source serves as an approximation to the thermal impact resulted from electrical state change, but eliminating the dependence of heat source on temperature and species. Thus, the τ_h can provide a reasonable approximation for the thermal response time after electrical variation.

In the revised manuscript, we further discuss the 'ambiguous situation' of OCV and TNV and justify our approach of using step change of heat source to evaluate the heat transfer characteristic time. (Supplementary Method, Lines 45-64 and Supplementary Figure 2)

"This section explains the strategy applied to investigate the thermal responses of SOC. ... Supplementary Fig. 2a-b present the responses of two identical SOECs, initially at 1.1V, after rapid voltage changes to OCV ($\approx 0.9V$) and TNV ($\approx 1.35V$), respectively. Even though the initial and final thermal states of both SOECs are identical, their temperature relaxation processes vary due to the distinct responses of current and heat source. This example highlights the complex relationship between heat source, current, and voltage, and suggests that manipulating the electrical state is not a straightforward approach for thermal investigation of SOCs. The heat source variation is, in fact, the primary driver of temperature change. In order to clearly define the characteristic time of thermal response, we manipulated the heat source rather than the electrical conditions in the thermal investigation. ... The step change of S_H^{total}

provides a reasonable approximation to the actual response of the heat source after electrical variation.”

L5: Please start sentence with “Employment of a” ...

This has been revised in the manuscript highlights:

“Employment of a 3-D numerical model to investigate transient behavior in Solid Oxide Cells (SOCs).”

L6: Please start sentence with “Quantification of the relationship...”

This has been revised in the manuscript highlights:

“Quantification of the relationship between SOC transients and key parameters.”

L7: Please start sentence with “Identification of the characteristic times... ”

This has been revised in the manuscript highlights:

“Identification of the characteristic times...”

L8: Please start sentence with “Validation of characteristic times against ... ”

This has been revised in the manuscript highlights:

“Validation of characteristic times against experiment and literature, demonstrating generality.”

L9: Please start sentence with “Elucidation of the potential of identified characteristic times for SOC design and control optimization with minimal computational cost”

This has been revised in the manuscript highlights:

“Elucidation of the potential of identified characteristic times for SOC design and control optimization with minimal computational cost.”

L36: please replace “store” with “convert” because SOEC “convert” electrical and thermal energies to chemical energy of produced hydrogen, carbon monoxide or syngas which can then be “stored”. Thus SOECs only enable storage or convert

This has been revised in the manuscript (Lines 25-26):

“...solid oxide electrolysis cells (SOECs) can convert the intermittent solar and wind energy to the chemical energy of alternative fuels such as hydrogen and syngas...”

L45: “Nerat [19] recommended increasing the anode supporting layer thickness to 0.1 mm”

This is not correct!

Nerat finds that 0.1mm is prone to starvation. This means he recommends maintaining the anode support thickness above 0.1mm

This does not mean to increase the thickness to at least 0.1mm, because support layers of technical or state of the art anode supported cells are 0.3mm and above, down from 1.5mm, to 1mm, to 0.5mm to now 0.3mm over the years. To the best of my knowledge, there is no technically relevant Anode Supported Cell, with a support thickness lower than 0.1mm. Please

reformulate the sentence.

Thanks much for pointing it out. We have revised the manuscript accordingly (Line 35):

“Nerat recommended maintaining the anode supporting layer thickness above 0.1 mm to prevent local fuel starvation...”

L67: What do the authors mean with the phrase “the incorporation of advanced heat and mass transfer theory within SOCs may stimulate new comprehension”. SOC literature is filled with heat and mass transfer theories. How advanced are these theories employed in this article.

In our view, the dimensional analysis is an advanced technology in the heat and mass transfer theory. Dimensional analysis offers the advantage of deriving generalized conclusions which are applicable to a broader range of cases. However, we observed that its application to heat and mass transfer is infrequently seen in SOC literature. Therefore, we have refined our statement in the revised manuscript (Lines 56-57):

“the incorporation of dimensional analysis within SOCs may stimulate new comprehension...”

L68 – L70: I do not see the analogy in structure and flow phenomenon between group I consisting of SOC, PEMFC, PCEC, i.e. protonic ceramic fuel cell/electrolyzer and group II consisting of alkaline fuel cells and electrolyser and flow battery. Please either elaborate more or stick to group I systems.

We thank the reviewer for providing the professional suggestion on the scope of the paper. We are indeed focusing on the discussions on group I (SOC, PEMFC, and PCEC). This has been revised in the manuscript (Lines 58-59):

“Moreover, the incorporation of dimensional analysis within SOCs may stimulate new comprehension of transient behavior in electrochemical cells exhibiting analogous structures and flow phenomena. Such cells encompass proton exchange membrane fuel cells (PEMFCs) and protonic ceramic fuel cells/electrolyzers.”

Revised main text (Lines 166-167):

“...indicate a potential for characterizing the dynamic processes of electrochemical cells with similar cell structures and flow phenomena to SOCs, such as PEMFCs and protonic ceramic fuel cells/electrolyzers.”

L85: please replace “water” with “steam” because at the SOC operation temperatures, it is most definitely steam that is produced and not water

This has been revised in the manuscript (Line 62):

“hydrogen and oxygen are consumed to produce steam and electricity.”

L89: please replace “voids” with “pores”

This has been revised in the manuscript (Line 66):

“...contain pores for gas transport.”

L90: please replace “exclusively” with “predominantly”, because in SOCs with not so high performance, the penetration depth goes beyond the generally < 10 μm thick functional layers

into the diffusion layers. This means the DLs have some degree of ionic conduction. This holds especially for the fuel electrodes, since some oxygen electrodes may have very high electronic conductive DL e.g. LSM but minimal ionic conductivity.

This has been revised in the manuscript (Line 66):

“...oxygen ion transport occurs predominantly within the solid oxide electrolyte, AFL, and CFL.”

L91: please include “or double-phase” after “triple-phase” because in Ni/GDC fuel electrodes for instance, conversion also take place at the pore/GDC double phase boundary due to mixed conduction of GDC under reducing atmosphere

This has been revised in the manuscript (Line 68):

“...where cathodic reactions take place at the triple-phase or double-phase boundaries.”

L187: Please replace “While both of the diffusive and convective time constants are inherently used to describe 1-D local transport phenomena. The time constants for 1-D transport ...” with “While both diffusive and convective time constants are inherently used to describe 1-D local transport phenomena, the time constants for 1-D transport ...”

This has been revised in the manuscript (Lines 97-98).

“While both diffusive and convective time constants are inherently used to describe 1-D local transport phenomena, the time constants for 1-D transport ...”

L197 and L208: the relaxation time of voltage is scaled with τ_m of fuel electrode and nicely shown that electrical response is limited by species transport in the fuel electrode. For sake of completeness, the authors should please include a scaling of the voltage response to τ_m of the air electrode for both SOEC and SOFC modes in corresponding figures. It should come out, that species transport in the air electrode is not limiting the electrical response. Otherwise, the authors should at least state that this was done and a value obtained (please state value) depicting the non-limitation of electrical response by species transport in the air electrode.

The statement of the τ_m^{air} has been supplemented in the manuscript.

Revised main text (lines 116-118):

“Interestingly, if using τ_m^{air} as time scale, the relaxation time of U^ would span a range from 2 to 20 τ_m^{air} (see Supplementary Fig. 4). This suggests that the species transport in the fuel electrode is the limiting factor of the electrical response for SOCs.”*

Supplementary Figure 4: ... When considering $\tau_{m,air}$ as the time scale, the relaxation time of U^* cannot converge to $1\tau_{m,air}$.

L212: Please remove “diffusivity of gas” and “material properties” from sentence. I doubt that “temperature exerts a profound impact” on diffusivity of gas and microstructure properties. Furthermore, from the cited reference, Reference 39, temperature impact on gas diffusivity in both electrodes is shown to be very small in both SOFC and SOEC between 750°C and 850°C compared to impact on activation overpotential

This has been revised in the manuscript.

Revised main text (lines 122-123):

“Temperature exerts a profound impact on SOC performance by affecting various properties such as the ionic conductivity of the electrolyte and the activation loss of electrochemical reactions”

L216: please replace “vice versa” with “beyond it”, as use of vice versa as such is rather unfortunate

This has been revised in the manuscript.

Revised main text (line 125):

“During the operation of SOEC, electrochemical reactions are endothermic below the thermal neutral voltage and exothermic beyond it.”

Table 5: Not clear which response variables belong to which Cell type, in the case of multiple response variables; Please either include table lines of use brackets or larger spacing for to distinguish Cell type

The Table 3 (formerly Table 5) has been improved according to the reviewer’s comment.

L260: Please replace “minor” with “lower”, because from figure 6, fuel electrode DL contributes 27.2% of void volume. Given that fuel and air channels contribute 35.8%, I would not call the contribution of at least the fuel electrode DL as “minor”. This can be employed for FLs or air electrode DL with percentages < 1 ;

This has been revised in the manuscript.

Revised main text (line 171):

“Due to their lower contributions to the void volume...”

Reviewer #2 (Remarks to the Author):

The methodology employed in this study is appropriately chosen, and the presented results are indeed remarkable. However, prior to considering the publication of this paper in this journal, I have outlined some comments to further enhance its quality:

1. It is suggested that in the concluding paragraph of the introduction section, a clear and comprehensive explanation of the novelty of your work and its primary objectives be provided.

We have revised the concluding paragraph accordingly to emphasize the novelty of our work

and its objectives.

2. A notable revision is required for the abstract to render it more specific and precise. Currently, it lacks the necessary level of detail required to effectively summarize the research

We have revised the abstract accordingly. In addition, we also shortened the abstract to 150 words to meet the format requirement of the journal.

3. In the realm of mass transfer, it might be worth considering the incorporation of non-dimensional numbers such as the Sherwood number and Reynolds number. Similarly, for the heat transfer aspect, the inclusion of the Nusselt number could potentially offer deeper insights. Could you please elucidate the rationale behind the decision to not employ these non-dimensional numbers in your work?

Thank you for your insightful comment. We appreciate your suggestion regarding the use of traditional dimensionless numbers such as the Sherwood, Reynolds, and Nusselt numbers. Indeed, these have been foundational in the field of mass and heat transfer.

However, in our study, we opted for an alternative approach by using time constant ratios, as presented in Table 4 (formerly Table 1). We believe this method offers a unique perspective as it directly compares the speed of various physical processes, which aligns more closely with the primary focus of this paper - the investigation of heat and mass transfer rates.

To elaborate, the Reynolds number (Re), for instance, represents the ratio between inertial force and viscous force ($\frac{VL}{\nu}$). Alternatively, Re can be written as $\frac{L^2/\nu}{L/V}$, which directly compares how fast are the momentum diffusion and convection in L direction. This interpretation, in our view, is more insightful for understanding the transient characteristics of a physical process. Additionally, it's worth mentioning that the use of time constant ratios is not unprecedented, Ref [3] has also utilized this method.

Therefore, while we acknowledge the value of traditional dimensionless numbers, we find the ratios of time constants provide a more intuitive reflection of the speed of physics, which is central to the goals of our research. We hope this clarifies our choice of method. Please feel free to share any further thoughts or concerns.

4. It's of utmost importance to address whether the potential alteration of flow properties due to electrochemical reactions within the channel has been integrated into your model. This consideration holds the potential to impact the residence time for mass transport. A clarification regarding the inclusion of this phenomenon in your manuscript is necessary.

7. A comprehensive description of all properties and formulations integrated into your numerical model is imperative. This encompasses aspects such as effective diffusion, membrane properties, and the like.

Concerning the fluid properties, our numerical model modifies fluid properties in response to changing flow conditions like fluid temperature. This has been clarified in the revised manuscript (Lines 234-241):

"In our model, Fick's law is utilized to model gas diffusion within fluid channels, a process

inherently influenced by variations in temperature and pressure. The effective properties of porous media are determined by averaging the fluid and solid properties volumetrically.”

However, in the dimensional analysis, fluid properties are assumed to remain constant in dimensional analysis, allowing properties such as density ρ , diffusivity D , thermal conductivity k , and specific heat c_p to be taken out of the partial differential terms. This assumption and the Table 4 have been improved for further clarity.

5. Given the utilization of a numerical method, it would be advantageous to visually represent the mesh independence through a graph or table based on non-dimensional parameters like the Sherwood number and Nusselt number.

In our study, we used current and voltage as indicators for the mesh independence test to be consistent with other SOC literatures. Besides, the current and voltage are sensitive to temperature and species concentration, and therefore, effectively reflect the dependence of numerical results on mesh size. Although our mesh independence test is not included in the current manuscript, it has been given in our previous work [1]. The detailed test results are reproduced as follows:

Table: Mesh independence test.

Mesh number	Voltage [V]	Current density [A/cm ²]
2.9×10^5	1.20	-2.5132
5.5×10^5	1.20	-2.5280
1.0×10^6 (chosen)	1.20	-2.5387
1.5×10^6	1.20	-2.5410
2.3×10^6	1.20	-2.5397

Figure: Mesh independence test.

6. Additionally, kindly provide information regarding the step size utilized in your numerical method.

Our time-stepping strategy is designed to control the rate of change in key parameters. For instance, it limits the temperature variation to approximately 0.1 K/step and the H₂ mass fraction variation to around 0.2%/step. At the initial phase of voltage change, the time step size is minimized and can reach as low as 10⁻⁵s. Towards the end of the simulation, the time step size is increased, potentially up to 500s. Details of this approach has been explained in our previous work [1].

In the revised manuscript (Lines 244-246), we have improved the description of the time stepping, while more details can be found from our previous work [1].

“In addition, an adaptive time-stepping algorithm is employed to guarantee adequate temporal resolution for capturing electrical behaviors during the transient simulation. The time-step size at each step is constrained by the variation rates of species concentration and temperature.”

8. It is recommended to provide detailed explanations of membrane properties and other pertinent information, substantiated with appropriate references.

Thanks for the reviewer’s advice. The properties information has been provided in the Table A.1 in our previous work [1].

Table A.1
Material parameters used in the governing equations.

Parameters		Value	Unit	Ref.
Ionic conductivity	$\tilde{\sigma}_{\text{CFL}}$	2.8	S/m	[34]
	$\tilde{\sigma}_{\text{AFL}}$	5.0	S/m	[34]
	$\tilde{\sigma}_{\text{E}}^{\text{a}}$	$\delta_{\text{E}} \cdot \frac{4.19 \times 10^{12}}{T} e^{-\frac{RT}{90310}}$	S/m	[35]
Electronic conductivity	$\sigma_{\text{Int.}}$	870000	S/m	[34]
	σ_{CDL}	1773200	S/m	[34]
	σ_{CFL}	2288000	S/m	[34]
	σ_{ADL}	7300	S/m	[34]
	σ_{AFL}	3650	S/m	[34]
Thermal conductivity	$k_{\text{Int.}}$	20	W/m K	[34]
	k_{CDL}	6	W/m K	[34]
	k_{CFL}	6	W/m K	[34]
	k_{ADL}	2	W/m K	[34]
	k_{AFL}	2	W/m K	[34]
	k_{E}	2	W/m K	[34]
Porosity	ϵ_{CDL}	0.38	–	[34]
	ϵ_{CFL}	0.2	–	[34]
	ϵ_{ADL}	0.27	–	[34]
	ϵ_{AFL}	0.27	–	[34]
Density	ρ_{CDL}	3310	kg/m ³	[16]
	ρ_{CFL}	3310	kg/m ³	[16]
	ρ_{ADL}	3030	kg/m ³	[16]
	ρ_{AFL}	3030	kg/m ³	[16]
	ρ_{E}	5160	kg/m ³	[16]
	$\rho_{\text{Int.}}$	3030	kg/m ³	[16]
Thermal capacity	$c_{p,\text{CDL}}$	450	J/kg K	[16]
	$c_{p,\text{CFL}}$	450	J/kg K	[16]
	$c_{p,\text{ADL}}$	430	J/kg K	[16]
	$c_{p,\text{AFL}}$	430	J/kg K	[16]
	$c_{p,\text{E}}$	470	J/kg K	[16]
	$c_{p,\text{Int.}}$	550	J/kg K	[16]

^aThe ionic conductivity of electrolyte is calculated based on the overall ohmic resistance measured by Njodzefon et al. [35].

9.The omission of mentioning the limiting current in your manuscript is noted. Given its significance in the upscaling process, this parameter warrants inclusion and discussion.

We appreciate the reviewer's insightful comment regarding the limiting current. The limiting current is the theoretical maximum current density achievable under a determined reactant supply. In the revised manuscript, we have clarified the limiting current of each case in Table 1. Additionally, we have provided clarification in the simulation procedure (Lines 82-83) that the currents used in all the cases are within the limiting current.

“For all the cases, $i_{t=0}$ and $i_{t>0}$ are within the limiting current i_{lim} , which serves as the theoretical maximum current density achievable under a given reactant supply.”

10.The conclusion section necessitates revision in order to render your findings and their implications more transparent and concise.

Thanks for your comment. We have revised the conclusion part (now the section is named as “Discussion” per the journal’s formatting requirement), please see Lines 208-222.

By thoughtfully addressing these recommendations, your paper stands to significantly elevate its value and impact.

Thanks very much for your positive comments, we have carefully revised our manuscript according to your constructive comments.

Reference

- [1] Liang, Z., Wang, J., Wang, Y., Ni, M., & Li, M. (2023). Transient characteristics of a solid oxide electrolysis cell under different voltage ramps : Transport phenomena behind overshoots. *Energy Conversion and Management*, 279(January), 116759.
- [2] Resch, E., & Pharoah, J. G. (2008). Numerical and experimental characterisation of convective transport in solid oxide fuel cells.
- [3] Costa, V. A. F. (2002). A time scale-based analysis of the laminar convective phenomena. *International Journal of Thermal Sciences*, 41(12), 1131–1140. [https://doi.org/10.1016/S1290-0729\(02\)01399-6](https://doi.org/10.1016/S1290-0729(02)01399-6)

REVIEWER COMMENTS

Reviewer #1 (Remarks to the Author):

I would like to thank the authors for satisfactorily discussing all points I pointed out.

However, there are two more things to clarify:

(I): Line 124: "During the operation of SOEC, electrochemical reactions are endothermic below the thermal neutral voltage and exothermic beyond it. While under SOFC mode, electrochemical reactions are exothermic"

I think this sentence is misleading. The endothermicity of the electrochemical reactions is the result of all electrochemical reactions at the anode and cathode. It does not change before and beyond the TNV. Joule heat is always exothermic. However, before the TNV, the endothermicity of the electrochemical reactions dominates joule heat, such that the overall SOEC heat balance is endothermal before and exothermal after the TNV. Please correct the sentence accordingly.

(II) Maybe I am wrong, but with respect to supplementary figure 2, I would expect the heat sources at the cathode and anode to have opposing signs with the anode dominating due to the largest change in entropy of the anode. In supplementary figure 2, the heat source of the anode is larger than that of the cathode as expected, but I would have expected them to have opposing signs, such that their sum would then be equal and opposite in sign to that of the electrolyte, such that they cancel out at TNV.

Please verify and if possibly check the implication of this on the method.

Reviewer #3 (Remarks to the Author):

Discovering two general characteristic times of transient responses in solid oxide cells

The manuscripts discuss the transient characteristics in Solid Oxide Cells which is a very important issue. It should be noted that the SOFC is a complex multiphase system. The authors have done a good job when it comes to analysis but the manuscript needs serious organization. Please find major comments below:

- What are the assumptions made in this study?
- Introduction-Please write the novelty and the research gap in this study
- Abstract-It should include the major results and the take home message.
- Where are the conclusions?
- All equations should be properly cited.
- In Identifying key parameters by non-dimensional analysis: the authors should discuss and identify key parameters through the non-dimensional analysis for solid oxide fuel cell system. They should identify and discuss the relative importance of various factors. The authors should discuss their approach on how the results are important to measure/interpret and optimize the SOFC system
- Overall-Please follow a schematic approach in the paper to maintain a logical flow. Abstract-Introduction-Methodology (in detail) with assumptions, Results, Discussion and Conclusions. This means a major re-structure is required for the manuscript.
- Discuss sources of error and how to validate results

Reviewer #4 (Remarks to the Author):

Techniques to represent the transient behavior of electrochemical cells are usually computationally demanding. This work investigates the SOFC response during varying conditions. The authors conduct very relevant research. I only have some minor comments for final improvements.

- 1) Please add some quantitative data in the abstract section related to the main research-outcomes
- 2) L101.. The document can contain the presumptions that were utilized to apply conservation regulations to the control volume. It is preferable to include the considered cell's active region as well. It would be ideal for the first sections of the manuscript to provide some insight into the boundary conditions.
- 3) Is the analysis taking into account the effects of convection and Knudsen diffusion?
- 4) L235... Why is the extended Ficks law employed and the dusty gas model not preferred?
- 5) Physical interpretation of τ_M could be amended
- 6) The writers create a distinctive theoretical and empirical framework, and I think the introduction should emphasise this innovation a lot more. Research gaps should be communicated in a more direct manner, with a focus on the specific needs of the undertaken study.

7) It is well known that the design of flow field has significant influence on the transient response of SOFC. How to consider this factor in the work?

8) While it is true that transients occurring in PEMFC-like cells share similarities in shape, duration, and magnitude, how much of your assertion that this can be applied to other PEMFC-like electrochemical cells is authentic?

9) The conclusion can be improved to help the readers find the key information more convenient. The conclusion could be effectively used for highlighting the dispersed primary findings. Conclusion section is missing some perspective related to the future research work.

It is suggested that the work be published after carefully considering all of this feedback.

References

[1] Ali Aghaei, Javad Mahmoudimehr, Nima Amanifard, The impact of gas flow channel design on dynamic performance of a solid oxide fuel cell, *International Journal of Heat and Mass Transfer*, Volume 219, 2024, 124924, ISSN 0017-9310 <https://doi.org/10.1016/j.ijheatmasstransfer.2023.124924>

[2] Kavya VanajaRaghunath, Shaneeth Muliankeezhu & Aparna Kallingal, Transient response modeling of reactant concentration in polymer electrolyte membrane fuel cells during load change, *Energy Sources, Part A: Recovery, Utilization, and Environmental Effects*, 2021, <https://doi.org/10.1080/15567036.2021.1933266>.

[3] Bohan Li, Chaoyang Wang, Ming Liu, Jianlin Fan, Junjie Yan, Transient performance analysis of a solid oxide fuel cell during power regulations with different control strategies based on a 3D dynamic model, *Renewable Energy*, Volume 218, 2023, 119266, ISSN 0960-1481, <https://doi.org/10.1016/j.renene.2023.119266>

[4] E.A. El-Hay, M.A. El-Hameed, A.A. El-Fergany, Optimized Parameters of SOFC for steady state and transient simulations using interior search algorithm, *Energy*, Volume 166, 2019, Pages 451-461, ISSN 0360-5442, <https://doi.org/10.1016/j.energy.2018.10.038>.

Additional comments on previous reviewer's concern:

1) Do you feel that the novelty of the work and its primary objectives are clearly described?

Even though the motivation behind the work is covered in the introduction, a more 'direct link' to the novelty could be amended.

2) The authors have indicated that their use of time constant ratios (as presented in Table 4) provide a more intuitive reflection of the speed of physics in the model than using dimensionless quantities like Sherwood, Reynolds and Nusselt numbers. Do you agree with this statement for the present work?

Along with conduction and diffusion, the influence of convection will be taken into account while utilising the Sherwood and Nusselt numbers. The article must specify the degree to which convective transfer is incorporated into the work. Regarding the transients, which occur during load fluctuations, Figs. 2 and 3(c) provide a clear picture of the time constant ratios and their influence. Likewise, similar strategies have been used in literature. The writers provide experimental data to support their claims. To bolster the argument, a detailed explanation of how τ_m and τ_h dominate can be included.

3.) Are all properties and formulations integrated into the numerical model sufficiently described?

The primary text provides a description of the modeling's attributes and formulations. More information is included in the supplemental material.

4.) Are the inclusion of fluid properties in the numerical model clearly described, such as the potential alteration of flow properties due to

electrochemical reactions within the channel?

Flow channel variations and its effect on this need to be stated. Other aspects are taken into consideration.

5.) Is the limiting current sufficiently discussed in this work?

Yes, the authors have stated that the current considered lies within its theoretical limit. The I_{lim} values are also mentioned for each set of parameter variation. Discussion on this, based on generalization of the procedure, for varying power outputs/ stacking can be added for more clarity in this regard.

Following the resolution of these concerns, publication of the manuscript is encouraged.

Ref.: NCOMMS-23-30425B

Discovering two general characteristic times of transient responses in solid oxide cells

We would like to thank the editor and all reviewers for their constructive comments, which helped us improve our manuscript. We address each comment individually below. Changes made according to editor and reviewers' comments are highlighted in the revised manuscript.

Response to Comments by Reviewer #1:

I would like to thank the authors for satisfactorily discussing all points I pointed out.

However, there are two more things to clarify:

(I): Line 124: "During the operation of SOEC, electrochemical reactions are endothermic below the thermal neutral voltage and exothermic beyond it. While under SOFC mode, electrochemical reactions are exothermic"

I think this sentence is misleading. The endothermicity of the electrochemical reactions is the result of all electrochemical reactions at the anode and cathode. It does not change before and beyond the TNV. Joule heat is always exothermic. However, before the TNV, the endothermicity of the electrochemical reactions dominate joule heat, such that the overall SOEC heat balance is endothermal before and exothermal after the TNV. Please correct the sentence accordingly.

We are thankful for the reviewer's rigorous comment. The sentence has been revised as follows (Lines 130-131),

"Under SOEC mode, the cell is endothermic below the thermal neutral voltage and exothermic beyond it. While under SOFC mode, the cell is exothermic."

(II) Maybe I am wrong, but with respect to supplementary figure 2, I would expect the heat sources at the cathode and anode to have opposing signs with the anode dominating due to the largest change in entropy of the anode. In supplementary figure 2, the heat source of the anode is larger than that of the cathode as expected, but I would have expected them to have opposing signs, such that, their sum would then be equal and opposite in sign to that of the electrolyte, such that they cancel out at TNV.

Please verify and if possibly check the implication of this on the method.

We thank the reviewer for the insightful and professional comments. We agree with the reviewer that the heat sources at the two electrodes are different [4]. The heat sources presented in supplementary Figure 2 can be elucidated by the assumptions made in our numerical calculation of the heat source, as presented in our previous work [1].

The overall heat source of the two functional layers (FLs) is accurate. To prevent potential misunderstanding, we have revised the Supplementary Figure 2 to combine the heat sources of the two FLs, rather than presenting them separately for the AFL and CFL. As shown in the updated Supplementary Figure 2, the negative heat source in the FLs counterbalances the positive heat source in the electrolyte at TNV, which aligns with the reviewer's comment.

Supplementary Figure 2: Comparison of heat sources in different elements at 1.1V and TNV.

More detailed explanation of heat sources used in our numerical simulation is as follows:

In reality, the reactions occurring at the anode and cathode in SOEC are as follows:

The reversible heat source is induced by the changes of entropy:

$$S_{\text{rev}} = T(\Delta\tilde{S}_{\text{cat}}^{\text{real}} + \Delta\tilde{S}_{\text{an}}^{\text{real}})$$

where, the entropy changes for the cathodic and anodic reactions are given by:

$$\Delta\tilde{S}_{\text{cat}}^{\text{real}} = \tilde{S}_{\text{H}_2} + \tilde{S}_{\text{O}^{2-}} - \tilde{S}_{\text{H}_2\text{O}} - 2\tilde{S}_{\text{e}^-}$$

$$\Delta\tilde{S}_{\text{an}}^{\text{real}} = 0.5\tilde{S}_{\text{O}_2} - \tilde{S}_{\text{O}^{2-}} + 2\tilde{S}_{\text{e}^-}$$

where, the values of \tilde{S}_{H_2} , $\tilde{S}_{\text{H}_2\text{O}}$, and \tilde{S}_{O_2} can be obtained from thermodynamic property tables, while the transported entropies for an oxygen ion, $\tilde{S}_{\text{O}^{2-}}$, and an electron, \tilde{S}_{e^-} , are not readily available. This lack of data poses a challenge in determining the precise entropy change and reversible heat source within the cathode or anode functional layers.

However, our model is not aiming to resolve the heat source in each functional layer. Instead, it considers the overall thermal effects from both layers. As indicated in the Supplementary Information (lines 69-73):

“...the AFL, electrolyte, and CFL are adjacent and exhibit minimal thicknesses, approximately 0.01mm. Consequently, the temperatures across the AFL, electrolyte, and CFL are nearly uniform in the thickness direction. This uniformity allows for the thermal effects of AFL, CFL, and electrolyte to be approximated as a lumped heat source from a thin region.”

The overall heat source in AFL-electrolyte-CFL is critical for the thermal simulation, while the fraction of heat source assigned to each FL has little effect on the final solution [2].

Thus, it is feasible to compute the total entropy change of the overall reaction,

$$\Delta\tilde{S}_{\text{overall}} = \tilde{S}_{\text{H}_2} + 0.5\tilde{S}_{\text{O}_2} - \tilde{S}_{\text{H}_2\text{O}}$$

Then, a common approach in numerical calculations is to equally assign $\Delta\tilde{S}_{\text{overall}}$ to the anode and cathode functional layers [1,2]:

$$S_{\text{rev}}^{\text{AFL}} = S_{\text{rev}}^{\text{CFL}} = \frac{1}{2}T\Delta\tilde{S}_{\text{overall}}$$

This method is adapted in our study to calculate the reversible heat sources of AFL and CFL.

In this way, while the heat source value of each functional layer may be subject to uncertainties -- as noted by the reviewer -- the overall heat source of the two layers is accurate. This ensures the reliability of our simulation in capturing the thermal effects due to entropy changes. Therefore, we believe **our treatment on the heat source will not influence the research outcome of this study.**

Response to Comments by Reviewer #3

Discovering two general characteristic times of transient responses in solid oxide cells
The manuscripts discuss the transient characteristics in Solid Oxide Cells which is a very important issue. It should be noted that the SOFC is complex multiphase system. The authors have done a good job when it comes to analysis but the manuscript needs serious organization. Please find major comments below:

- Introduction-Please write the novelty and the research gap in this study

We thank the reviewer's comment. The introduction part has been improved to emphasize the research gap and novelty, as presented in lines 51-59:

"...The absence of a generalized expression necessitates substantial computational and experimental efforts to characterize the transient behaviors of SOCs. This research gap hinders the optimizations of SOC design and control strategies, as well as the application of SOCs in renewable energy storage and conversion.

To address the prevailing research gap, the present study develops a distinctive theoretical framework for the investigation of transient behaviors of SOCs. We uncover generalized and straightforward mathematical formulas to calculate characteristic times that govern the transport phenomena within the SOCs. Our findings offer innovative insights into the correlation between transient characteristics of SOCs and various parameters, thereby enabling more effective design and control of SOCs at greatly reduced computational and experimental efforts."

- Abstract-It should include the major results and the take home message.

We appreciate the reviewer's constructive comment. We have improved the abstract accordingly.

"...Through comprehensive numerical analysis, we find that the thermal and gaseous response times of SOCs upon rapid electrical variations are at the order of two characteristic times (τ_h and τ_m), respectively. The gaseous response time is approximately $1\tau_m$, and the thermal response time aligns with roughly $2\tau_h$. These characteristic times represent the overall heat and mass transfer rates within the cell, and their mathematical relationships with various SOC design and operating parameters are revealed. Validation of τ_h and τ_m is achieved through comparison with an in-house experiment and existing literature data, achieving the same order of magnitude for a wide range of electrochemical cells, showcasing their potential use for characterizing transient behaviors in a wide range of electrochemical cells. ..."

- All equations should be properly cited.

In the revised manuscript, citations are supplemented for the equations in Table 4 and Supplementary Eq. (12).

- What are the assumptions made in this study?
- Where are the conclusions?
- Overall-Please follow a schematic approach in the paper to maintain a logical flow. Abstract-Introduction-Methodology (in detail) with assumptions, Results, Discussion and Conclusions. This means a major re-structure is required for the manuscript.

We appreciate the reviewer's constructive comment. When firstly submitted to the journal, our manuscript was following the structure of 'Abstract-Introduction-Methodology-Results-Discussion-Conclusions'. But during the revision process, we have revised the structure to – 'Abstract-Introduction-Results-Discussion (optional)-Methods (optional)' according to the journal's formatting requirement. Thus, the assumptions made for the numerical modeling and theoretical derivations are listed in the Methods section. A conclusive summary is arranged into the Discussion section.

- In Identifying key parameters by non-dimensional analysis: the authors should discuss and identify key parameters through the non-dimensional analysis for solid oxide fuel cell system. They should identify and discuss the relative importance of various factors. The authors should discuss their approach on how the results are important to measure/interpret and optimize the SOFC system

We appreciate the reviewer's constructive comment. In the section of 'Identifying key parameters by non-dimensional analysis', we employ non-dimensional analysis to find the factors that influence SOC transients. To evaluate the importance of these factors, we conduct a parametric study using numerical simulations, the results of which are presented in Tables 1 and 2, as well as in Figure 2. In Figure 3a and the section of 'Application scenario: guidance of SOC control', we quantitatively identify the relative importance of the influential factors and demonstrate how the characteristic times can be utilized for SOC design. Furthermore, in the Discussion section, we summarized the key methodology and findings.

- Discuss sources of error and how to validate results

We thank the reviewer for the comment. The sources of error in the numerical model include the Butler–Volmer (B–V) equation and the extended Fick's law. The B–V equation, which is deduced from single-step and single-electron-transfer reactions, is adopted to model the multi-electron and multi-step reactions in SOCs. The use of B-V equation may introduce errors when calculating electrochemical reaction rates. Besides, as our numerical model does not account for the realistic microstructures of electrodes, the microstructural effects on the gas diffusion is considered through the extended Fick's law, which may lead to modeling errors. In our previous work [1], we validated the numerical model by comparing the simulation results with the experimental results for both electrolysis and fuel cell modes in different temperatures and inlet conditions. The validation results suggest that the numerical model has sufficient accuracy for this study.

Validation of the numerical model in our previous work [1].

We have revised the manuscript accordingly (lines 244 - 248):

“In porous media, the realistic microstructures of electrodes are neglected, and the microstructural and Knudsen effects on mass diffusion are considered through the extended Fick’s law \cite{njodzefon2013electrochemical} ...”

“Electrochemical reactions are modeled by employing the Butler-Volmer equation, which is derived from single-step and single-electron-transfer reactions and shows sufficient accuracy when calculating the multi-electron and multi-step reactions in SOCs \cite{LIANG2023116759,njodzefon2013electrochemical} ...”

Response to Comments by Reviewer #4

Techniques to represent the transient behavior of electrochemical cells are usually computationally demanding. This work investigates the SOFC response during varying conditions. The authors conduct very relevant research. I only have some minor comments for final improvements.

1) Please add some quantitative data in the abstract section related to the main research-outcomes

We are thankful for the reviewer’s comment. Our work claims that the characteristic time can predict the order of magnitude of response time rather than exact values. We have revised the abstract accordingly quantitative,

“Through comprehensive numerical analysis, we find that the thermal and gaseous response times of SOCs upon rapid electrical variations are at the order of two characteristic times (τ_h and τ_m), respectively. The gaseous response time is approximately $1\tau_m$, and the thermal response time aligns with roughly $2\tau_h$...”

“Validation of τ_h and τ_m is achieved through comparison with an in-house experiment and existing literature data, achieving the same order of magnitude for a wide range of electrochemical cells, ...”

2) L101.. The document can contain the presumptions that were utilized to apply conservation

regulations to the control volume. It is preferable to include the considered cell's active region as well. It would be ideal for the first sections of the manuscript to provide some insight into the boundary conditions.

Thank you for the comment. The active regions of the cell, i.e., the functional layers (FL), have been considered when applying conservation laws to the control volume. While calculating the characteristic times using Eq. (1) and (2), we have considered geometrical parameters (porosity ϵ_{FL} and thickness δ_{FL}) and thermal properties (specific heat $c_{p,CFL}$ and $c_{p,AFL}$) of the FLs.

In addition, we have added additional description to the presumptions for control volume and boundary conditions in the manuscript.

(Lines 102-107)

“...we apply the concept of control volume from Fluid Mechanics to the SOC. Control volume is a closed region defined in space, which is utilized to focus our attention on the mass and energy crossing the boundary and the conservation law within the region. Either the fuel side or air side of the SOC can be considered as a control volume. Take the fuel side as an example, the inlet and outlet of fuel channel allow the crossing of fuel flow, while the other boundaries of the control volume are impermeable. The total mass of the control volume obeys the conservation law.”

(Lines 149-154)

“...the control volume concept with energy conservation principle is applied (see Supplementary Methods for detailed derivation). Given the boundary conditions of the system under investigation (Table 5), there are only two inlets and two outlets that exchange heat with the environment, while the other boundaries are adiabatic. Thus, the total inflow heat transfer rate of the system can be represented by the sum of the enthalpy of fluids at the two inlets. Subsequently, the characteristic time of heat transfer within SOCs, denoted as, τ_h , was derived as follows,...”

3) Is the analysis taking into account the effects of convection and Knudsen diffusion?

Thank you for the comment. The numerical model considers the convection term and Knudsen diffusion for the porous flow. In our model, the calculation of the effective diffusivity in porous media follows Njodzefon et al.'s work [3], which considers the Knudsen effect. The Table 3 in our previous work [1] listed the governing equations of the flow in porous media includes the convection terms (listed below).

Porous media (ADL, AFL, CFL, CDL)	
Continuity	$\frac{\partial(\epsilon\rho)}{\partial t} + \nabla \cdot (\rho\vec{v}) = S_m$
Momentum ^{a,c}	$\frac{\partial}{\partial t}(\rho\vec{v}) + \nabla \cdot (\rho\vec{v}\vec{v}) = -\nabla p + \nabla \cdot (\vec{\tau}) - \frac{\mu}{k} \vec{v}$
Species	Convection terms $\frac{\partial}{\partial t}(\epsilon\rho Y_i) + \nabla \cdot (\rho\vec{v}Y_i) + \nabla \cdot \vec{J}_i = S_i$
Energy ^b	$\frac{\partial}{\partial t}(\epsilon\rho_f E_f + (1-\epsilon)\rho_s E_s) + \nabla \cdot (\vec{v}(\rho_f E_f + p))$ $= \nabla \cdot [(\epsilon k_f + (1-\epsilon)k_s)\nabla T - (\sum_i h_i \vec{J}_i)] + S_h$
State function	$\rho = \frac{pM}{RT}$ (ideal gas)
Electronic charge	$\nabla \cdot (\sigma \nabla \phi_{cle}) + S_{cle} = 0$
Ionic charge	$\nabla \cdot (\tilde{\sigma} \nabla \phi_{ion}) + S_{ion} = 0$

To clarify this, lines 244-245 in the manuscript has been revised as follows:

“In porous media, the convection of momentum, mass, and heat is considered, and the extended Fick’s law [33] is used to incorporate the microstructural and Knudsen effects on mass diffusivity.”

4) L235... Why is the extended Ficks law employed and the dusty gas model not preferred?

We agree with the reviewer that the dusty gas model is usually more suitable for simulating the multicomponent gas diffusion through porous media. In the present study, only 2 gas species exist in both anode and cathode. Thus, the extended Fick’s law can be used for simulating the gas diffusion through the porous electrodes with acceptable accuracy. Besides, as demonstrated in our previous work, our 3-D model is developed based on Njodzefon et al. [3]’s 1-D model and experimental data. The use of extended Fick’s law, consistent with Njodzefon’s work [3], has achieved satisfactory accuracy. But thanks very much for the reviewer’s comment, we will consider the dusty gas model in the future when diffusion of more than 3 gas species take place in the porous electrodes.

5) Physical interpretation of τ_M could be amended

We appreciate the reviewer's suggestion. This has been improved in the Lines 109-110:

“...reflects the time required for the fluid to flow through the void volume in SOC.”

6) The writers create a distinctive theoretical and empirical framework, and I think the introduction should emphasise this innovation a lot more. Research gaps should be communicated in a more direct manner, with a focus on the specific needs of the undertaken study.

We appreciate the reviewer's constructive suggestions. This has been improved in lines 51-59:

“...The absence of a generalized expression necessitates substantial computational and experimental efforts to characterize the transient behaviors of SOCs. This research gap hinders the optimizations of SOC design and control strategies, as well as the application of SOCs in renewable energy storage and conversion.

To address the prevailing research gap, the present study develops a distinctive theoretical framework for the investigation of transient behaviors of SOCs. We uncover generalized and straightforward mathematical formulas to calculate characteristic times that govern the transport phenomena within the SOCs. Our findings offer innovative insights into the correlation between transient characteristics of SOCs and various parameters, thereby enabling more effective design and control of SOCs at greatly reduced computational and experimental efforts.”

7) It is well known that the design of flow field has significant influence on the transient response of SOFC. How to consider this factor in the work?[Additional comments on previous reviewer’s concern]

4.) Are the inclusion of fluid properties in the numerical model clearly described, such as the potential alteration of flow properties due to electrochemical reactions within the channel?

Flow channel variations and its effect on this need to be stated. Other aspects are taken into consideration.

We appreciate the reviewer's insightful comment. Our work considers the global effects of channel design on SOC transients. For example, Eq.(1) considers the width (W), length (L), and height (H) of channel while calculating the mass-transfer characteristic time, τ_m . Besides, Table 1 shows that the channels of Case 1, Case 5, Case 6, Case 22, and Case 23 have various width-to-height ratios. The response times of these cases are at the order of $1\tau_m$, which suggests that the proposed characteristic time can effectively estimate the transients of SOCs with different channel width-to-height ratios.

In literature, the channels of SOCs are not only limited in the rectangular type. In Table 3, we use the proposed characteristic time to predict the response times of SOCs with various channel designs, such as tubular design, flat-tube design, and cavity design (our in-house experiment). Despite the diversity in channel designs, our estimated response times remain the same order of magnitude when compared with the modeling and the experimental results from the literature and our in-house experiments. By careful analysis and validation, we are confident that our proposed characteristic times have a wide applicability and can give good estimates for cells with various channel designs.

We agree with the reviewer regarding the significant impact of channel design. We further clarify this part in the discussion section:

(Lines 227-231)

“In addition, the proposed characteristic times are intended to generalize the response times in various SOCs. To acquire the exact values of response times, the local effects of SOC designs, such as the cross-sectional shape of channel \cite{AGHAEI2024124924}, should be considered. For future work, it would be intriguing to compare the transient performance of different channel designs that share the same general characteristic times. In such research, the generalized characteristic time could serve as a benchmark to facilitate channel optimization.”

8) While it is true that transients occurring in PEMFC-like cells share similarities in shape, duration, and magnitude, how much of your assertion that this can be applied to other PEMFC-like electrochemical cells is authentic?

Thank you for this comment. In our manuscript, we assert that our theoretical framework may enhance comprehension of the transient responses observed in PEMFCs and protonic ceramic fuel cells (PCFCs). In Table 3, our validation results show that the mass-transfer characteristic time τ_m can be used to explain part of the transients in PEMFC. However, it should be noted that the mass-transfer physics in PEMFC are more complicated than SOC. For example, PEMFC involves liquid and gas phases, and the water transport in the membrane significantly influences electrical efficiency. Thus, one characteristic time may be insufficient to describe all the dynamic mass-transfer processes in PEMFC, and more investigation is needed. Our work can provide a new direction for these investigations.

In terms of PCFCs, their operating conditions and cell components closely resemble those of SOFCs. We believe that the proposed characteristic times can be directly applied to analyze the transients of PCFCs. However, given that the PCFC technology is less mature than that of SOFCs, there is a scarcity of studies on PCFC transients. To corroborate our theoretical

framework for PCFCs, there is a need for more experimental data.

9) The conclusion can be improved to help the readers find the key information more convenient. The conclusion could be effectively used for highlighting the dispersed primary findings. Conclusion section is missing some perspective related to the future research work.

We appreciate the reviewer's constructive suggestions. The conclusion and future research work has been further clarified in the revised manuscript in lines 213-231:

“In summary, we proposed general expressions of two characteristic times (τ_h and τ_m) to represent the overall heat and mass transfer rates within various SOCs under different operating conditions. The gaseous and thermal response times upon rapid electrical variations are at the order of τ_h and τ_m , respectively. The effectiveness of τ_h and τ_m was also validated against our numerical simulations and experiments, as well as the transient characteristics of several types of SOCs and a PEMFC reported in the literature. In terms of potential applications, on one hand, τ_h and τ_m can serve as quantifiable and easy-to-calculate indicators for designing SOCs with desired transient characteristics. On the other hand, τ_m can enhance the generalizability of existing SOC control strategies. The proposed characteristic times could enrich the theoretical comprehension of SOCs, especially during unstable operating conditions, and could potentially boost the development of SOC for renewable energy storage and conversion.

In terms of the methodology to derive the characteristic times, we adopted the non-dimensional analysis to identify the parameters and time constants that directly influence SOC transients. This methodology leads to an innovative theoretical framework to study the heat and mass transfer phenomena in SOCs, which may also spark new insights into the transient behavior of electrochemical cells with similar structures and flow phenomena, such as proton exchange membrane fuel cells/electrolyzers and protonic ceramic fuel cells/electrolyzers.

In addition, the proposed characteristic times are intended to generalize the response times in various SOCs. To acquire the exact values of response times, the local effects of SOC designs, such as the cross-sectional shape of channel \cite{AGHAEI2024124924}, should be considered. For future work, it would be intriguing to compare the transient performance of different channel designs that share the same characteristic times. In such research, the characteristic time could serve as a benchmark to facilitate channel optimization.”

It is suggested that the work be published after carefully considering all of this feedback.

References

[1] Ali Aghaei, Javad Mahmoudimehr, Nima Amanifard, The impact of gas flow channel design on dynamic performance of a solid oxide fuel cell, International Journal of Heat and Mass Transfer, Volume 219, 2024, 124924, ISSN 0017-9310
<https://doi.org/10.1016/j.ijheatmasstransfer.2023.124924>

[2] Kavya VanajaRaghunath, Shaneeth Muliankeezhu & Aparna Kallingal, Transient response

modeling of reactant concentration in polymer electrolyte membrane fuel cells during load change, *Energy Sources, Part A: Recovery, Utilization, and Environmental Effects*, 2021, <https://doi.org/10.1080/15567036.2021.1933266>.

[3] Bohan Li, Chaoyang Wang, Ming Liu, Jianlin Fan, Junjie Yan, Transient performance analysis of a solid oxide fuel cell during power regulations with different control strategies based on a 3D dynamic model, *Renewable Energy*, Volume 218, 2023, 119266, ISSN 0960-1481, <https://doi.org/10.1016/j.renene.2023.119266>

[4] E.A. El-Hay, M.A. El-Hameed, A.A. El-Fergany, Optimized Parameters of SOFC for steady state and transient simulations using interior search algorithm, *Energy*, Volume 166, 2019, Pages 451-461, ISSN 0360-5442, <https://doi.org/10.1016/j.energy.2018.10.038>.

Additional comments on previous reviewer's concern:

1) Do you feel that the novelty of the work and its primary objectives are clearly described? Even though the motivation behind the work is covered in the introduction, a more 'direct link' to the novelty could be amended.

We appreciate the reviewer's comments. The introduction in the manuscript has been improved, as shown in Lines 50-62:

"In summary, the existing research does not adequately describe the relationship between SOC transients and their associated design and operating parameters. The absence of a generalized expression necessitates substantial computational and experimental efforts to characterize the transient behaviors of SOCs. This research gap hinders the optimizations of SOC design and control strategies, as well as the application of SOCs in renewable energy storage and conversion.

To address the prevailing research gap, the present study develops a distinctive theoretical framework for the investigation of transient behaviors of SOCs. We uncover generalized and straightforward mathematical formulas to calculate characteristic times that govern the transport phenomena within the SOCs. Our findings offer innovative insights into the correlation between transient characteristics of SOCs and various parameters, thereby enabling more effective design and control of SOCs at greatly reduced computational and experimental efforts. Moreover, the incorporation of non-dimensional analysis within SOCs may stimulate new comprehension of transient behavior in electrochemical cells with analogous structures and flow phenomena. Such cells encompass proton exchange membrane fuel cells (PEMFCs) and protonic ceramic fuel cells/electrolyzers."

2) The authors have indicated that their use of time constant ratios (as presented in Table 4) provide a more intuitive reflection of the speed of physics in the model than using dimensionless quantities like Sherwood, Reynolds and Nusselt numbers. Do you agree with this statement for the present work?

Along with conduction and diffusion, the influence of convection will be taken into account while utilising the Sherwood and Nusselt numbers. The article must specify the degree to which convective transfer is incorporated into the work. Regarding the transients, which occur during load fluctuations, Figs. 2 and 3(c) provide a clear picture of the time constant ratios and their influence. Likewise, similar strategies have been used in literature. The writers provide experimental data to support their claims. To bolster the argument, a detailed explanation of how τ_m and τ_h dominate can be included.

We appreciate the reviewer's comments. Our numerical model incorporates convection for the heat and mass transfer in channels and porous media, as indicated in Table 4 of the manuscript. Besides, our non-dimensional analysis not only accounts for convection terms but also introduces the convective time constant, $L_{\text{cell}}/V_{\text{in}}$. Thus, our simulation and analysis incorporate the convective effects into equations in a straightforward way.

Table 4: Non-dimensional analysis of the heat and mass transfer processes in SOC.

Simplified governing equations for non-dimensional analysis	Domains
Conservation of species	
$\frac{\partial Y_i}{\partial t} + \vec{V} \cdot \nabla Y_i - D \nabla^2 Y_i = 0$	Fluid channel
$\frac{\partial(\varepsilon Y_i)}{\partial t} + \vec{V} \cdot \nabla Y_i - D_{\text{eff}} \nabla^2 Y_i = 0$	ADL,CDL
$\frac{\partial(\varepsilon Y_i)}{\partial t} + \vec{V} \cdot \nabla Y_i - D_{\text{eff}} \nabla^2 Y_i = \frac{S_{m,i}}{p}$	AFL,CFL
Conservation of energy	
$\frac{\partial T}{\partial t} + \vec{V} \cdot \nabla T - \alpha \nabla^2 T = 0$	Fluid channel
$\frac{\partial T}{\partial t} + \frac{\rho C_p}{\rho_{\text{cell}} C_p, \text{eff}} \vec{V} \cdot \nabla T - \alpha_{\text{eff}} \nabla^2 T = \frac{S_h}{\rho_{\text{cell}} C_p, \text{eff}}$	Porous ^d
$\frac{\partial T}{\partial t} - \alpha_s \nabla^2 T = \frac{S_h}{\rho_s C_p, s}$	Solid ^b
Scaling & definitions^c	
$t^* = \frac{t}{\tau_m}$ or $\frac{t}{\tau_h}$, $Y_i^* = \frac{Y_i - Y_{i, \text{ref}}}{\Delta Y_i} = \frac{Y_i - Y_{i, \text{ref}}}{Y_{i, \text{ref}} - Y_{i, \text{ref}}}$, $X_i^* = \frac{X_i - X_{i, \text{ref}}}{X_{i, \text{ref}} - X_{i, \text{ref}}}$, $V^* = \frac{V}{V_{\text{in}}}$, $T^* = \frac{T - T_{\text{ref}}}{\Delta T} = \frac{T - T_{\text{ref}}}{T_{\text{ref}} - T_{\text{ref}}}$, $\nabla^* = L_{\text{cell}} \nabla$, $x^* = \frac{x}{W_{\text{cell}}}$, $y^* = \frac{y}{L_{\text{cell}}}$, $z^* = \frac{z}{H_{\text{cell}}}$ or $\frac{z}{\delta_{\text{ch}}}$, or $\frac{z}{\delta_{\text{ch}}}$, $S_{m,i}^* = \frac{S_{m,i} \tau_m}{\rho_{\text{cell}} V_{\text{in}}}$, $S_h^* = \frac{S_h \tau_m}{\rho_{\text{cell}} C_p, \text{eff}} \Delta T$ or $\frac{S_h \tau_m}{\rho_s C_p, s} \Delta T$, $\alpha = \frac{k}{\rho C_p}$, $\alpha_{\text{eff}} = \frac{k_{\text{eff}}}{\rho_{\text{cell}} C_p, \text{eff}}$, $\alpha_s = \frac{k_s}{\rho_s C_p, s}$	
Dimensionless equations	Domains
Conservation of species	
$\frac{\partial Y_i^*}{\partial t^*} + \frac{\tau_m}{L_{\text{cell}} V_{\text{in}}} \vec{V}^* \cdot \nabla^* Y_i^* - \left(\frac{\tau_m}{W_{\text{cell}}^2 D} \frac{\partial^2 Y_i^*}{\partial x^{*2}} + \frac{\tau_m}{L_{\text{cell}}^2 D} \frac{\partial^2 Y_i^*}{\partial y^{*2}} + \frac{\tau_m}{H_{\text{cell}}^2 D} \frac{\partial^2 Y_i^*}{\partial z^{*2}} \right) = 0$	Fluid channel
$\varepsilon \frac{\partial Y_i^*}{\partial t^*} + \frac{\tau_m}{L_{\text{cell}} V_{\text{in}}} \vec{V}^* \cdot \nabla^* Y_i^* - \left(\frac{\tau_m}{W_{\text{cell}}^2 D_{\text{eff}}} \frac{\partial^2 Y_i^*}{\partial x^{*2}} + \frac{\tau_m}{L_{\text{cell}}^2 D_{\text{eff}}} \frac{\partial^2 Y_i^*}{\partial y^{*2}} + \frac{\tau_m}{H_{\text{cell}}^2 D_{\text{eff}}} \frac{\partial^2 Y_i^*}{\partial z^{*2}} \right) = 0$	ADL,CDL
$\varepsilon \frac{\partial Y_i^*}{\partial t^*} + \frac{\tau_m}{L_{\text{cell}} V_{\text{in}}} \vec{V}^* \cdot \nabla^* Y_i^* - \left(\frac{\tau_m}{W_{\text{cell}}^2 D_{\text{eff}}} \frac{\partial^2 Y_i^*}{\partial x^{*2}} + \frac{\tau_m}{L_{\text{cell}}^2 D_{\text{eff}}} \frac{\partial^2 Y_i^*}{\partial y^{*2}} + \frac{\tau_m}{\delta_{\text{ch}}^2 D_{\text{eff}}} \frac{\partial^2 Y_i^*}{\partial z^{*2}} \right) = S_{m,i}^*$	AFL,CFL
Conservation of energy	
$\frac{\partial T^*}{\partial t^*} + \frac{\tau_m}{L_{\text{cell}} V_{\text{in}}} \vec{V}^* \cdot \nabla^* T^* - \left(\frac{\tau_m}{W_{\text{cell}}^2 \alpha} \frac{\partial^2 T^*}{\partial x^{*2}} + \frac{\tau_m}{L_{\text{cell}}^2 \alpha} \frac{\partial^2 T^*}{\partial y^{*2}} + \frac{\tau_m}{H_{\text{cell}}^2 \alpha} \frac{\partial^2 T^*}{\partial z^{*2}} \right) = 0$	Fluid channel
$\frac{\partial T^*}{\partial t^*} + \frac{\tau_m}{L_{\text{cell}} V_{\text{in}}} \frac{\rho C_p}{\rho_{\text{cell}} C_p, \text{eff}} \vec{V}^* \cdot \nabla^* T^* - \left(\frac{\tau_m}{W_{\text{cell}}^2 \alpha_{\text{eff}}} \frac{\partial^2 T^*}{\partial x^{*2}} + \frac{\tau_m}{L_{\text{cell}}^2 \alpha_{\text{eff}}} \frac{\partial^2 T^*}{\partial y^{*2}} + \frac{\tau_m}{\delta_{\text{ch}}^2 \alpha_{\text{eff}}} \frac{\partial^2 T^*}{\partial z^{*2}} \right) = S_h^*$	ADL, CDL
$\frac{\partial T^*}{\partial t^*} + \frac{\tau_m}{L_{\text{cell}} V_{\text{in}}} \frac{\rho C_p}{\rho_s C_p, s} \vec{V}^* \cdot \nabla^* T^* - \left(\frac{\tau_m}{W_{\text{cell}}^2 \alpha_s} \frac{\partial^2 T^*}{\partial x^{*2}} + \frac{\tau_m}{L_{\text{cell}}^2 \alpha_s} \frac{\partial^2 T^*}{\partial y^{*2}} + \frac{\tau_m}{\delta_{\text{ch}}^2 \alpha_s} \frac{\partial^2 T^*}{\partial z^{*2}} \right) = S_h^*$	AFL, CFL
$\frac{\partial T^*}{\partial t^*} - \left(\frac{\tau_m}{W_{\text{cell}}^2 \alpha_s} \frac{\partial^2 T^*}{\partial x^{*2}} + \frac{\tau_m}{L_{\text{cell}}^2 \alpha_s} \frac{\partial^2 T^*}{\partial y^{*2}} + \frac{\tau_m}{\delta_{\text{ch}}^2 \alpha_s} \frac{\partial^2 T^*}{\partial z^{*2}} \right) = S_h^*$	Solid ^b

For the dominance of τ_m and τ_h , as they are the universal time scales in the governing equations, thus playing a dominating role in transient behaviors. But there is no straightforward way to estimate the values of the universal time scales, and this is the research gap this work will address. This has been clarified in the manuscript, lines 293-295.

"... τ_m or τ_h serves as the numerator of all time ratios, representing the global characteristic time of mass/heat transfer within the entire SOC. Therefore, the expressions of τ_m and τ_h are critical for generalizing SOC transient characteristics but their expressions remain unknown in non-dimensional analysis."

3.) Are all properties and formulations integrated into the numerical model sufficiently described?

The primary text provides a description of the modeling's attributes and formulations. More information is included in the supplemental material.

We appreciate the reviewer's explanation.

5.) Is the limiting current sufficiently discussed in this work?

Yes, the authors have stated that the current considered lies within its theoretical limit. The Ilim values are also mentioned for each set of parameter variation. Discussion on this, based on generalization of the procedure, for varying power outputs/ stacking can be added for more clarity in this regard.

We appreciate the reviewer' valuable comment. The varying power outputs have been added in Table 1. For stacking, although our numerical model is a single-channel model, but our validation was done on cells and stacks, as presented in Table 2.

Following the resolution of these concerns, publication of the manuscript is encouraged.

Reference

- [1] Liang, Z., Wang, J., Wang, Y., Ni, M., & Li, M. (2023). Transient characteristics of a solid oxide electrolysis cell under different voltage ramps : Transport phenomena behind overshoots. *Energy Conversion and Management*, 279(January), 116759.
- [2] Resch, E., & Pharoah, J. G. (2008). Numerical and experimental characterisation of convective transport in solid oxide fuel cells.
- [3] Njodzefon, J.-C., Klotz, D., Kromp, A., Weber, A., & Ivers-Tiffée, E. (2013). Electrochemical Modeling of the Current-Voltage Characteristics of an SOFC in Fuel Cell and Electrolyzer Operation Modes. *Journal of The Electrochemical Society*, 160(4), F313–F323. <https://doi.org/10.1149/2.018304jes>
- [4] Zheng, K., Sun, Q., & Ni, M. (2013). Local Non-Equilibrium Thermal Effects in Solid Oxide Fuel Cells with Various Fuels. *Energy Technology*, 1(1), 35-41.

REVIEWERS' COMMENTS

Reviewer #1 (Remarks to the Author):

I would like to thank the authors for addressing all the issues I raised to my satisfaction. The requested changes were understood and effected while a solid rebutal backing employed assumptions was made to my full satisfaction.

Thanks to the authors.

Reviewer #3 (Remarks to the Author):

The authors addressed all comments successfully. I recommend the acceptance of this article.

Reviewer #4 (Remarks to the Author):

The contributors have thoughtfully responded to the reviewer's critique. The discussion session illustrate the main assertions of the work, and the findings are noteworthy. No further queries/comments from my side.

Ref.: NCOMMS-23-30425C

Discovering two general characteristic times of transient responses in solid oxide cells

We are grateful to the editor and the reviewers for their insightful comments, which have significantly contributed to the improvement of our manuscript.

Response to Comments by Reviewer #1:

I would like to thank the authors for addressing all the issues I raised to my satisfaction. The requested changes were understood and effected while a solid rebutal backing employed assumptions was made to my full satisfaction.

Thanks to the authors.

We appreciate Reviewer #1's positive feedback and are pleased to hear that the revisions and explanations provided have met your expectations. Thank you for your thorough review and constructive suggestions, which have undoubtedly improved our manuscript.

Response to Comments by Reviewer #3:

The authors addressed all comments successfully. I recommend the acceptance of this article.

We thank Reviewer #3 for the positive evaluation and for recommending the acceptance of our article. We are grateful for your constructive comments, which guided our revisions.

Response to Comments by Reviewer #4:

The contributors have thoughtfully responded to the reviewer's critique. The discussion session illustrate the main assertions of the work, and the findings are noteworthy. No further queries/comments from my side.

We appreciate Reviewer #4's positive remarks. Your comments have been invaluable in strengthening our manuscript.